# EMBODIED-R1: REINFORCED EMBODIED REASONING FOR GENERAL ROBOTIC MANIPULATION

**Yifu Yuan**[1,†]**, Haiqin Cui**[1]**, Yaoting Huang**[1]**, Fei Ni**[1]**, Zibin Dong**[1]**, Pengyi Li**[1]
**Yan Zheng**[1]**, Hongyao Tang**[1]**, Jianye Hao**[1,*]
[1]College of Intelligence and Computing, Tianjin University
[†]Project Leader. [*]Corresponding authors: Jianye Hao (jianye.hao@tju.edu.cn).

## ABSTRACT

Generalization in embodied AI is hindered by the "seeing-to-doing gap", stemming from data scarcity and embodiment heterogeneity. To address this, we pioneer "pointing" as a unified, embodiment-agnostic intermediate representation, defining four core embodied pointing abilities that bridge high-level vision-language comprehension with low-level action primitives. We introduce **Embodied-R1**, a 3B Vision-Language Model (VLM) specifically designed for embodied reasoning and pointing. We use a wide range of embodied and general visual reasoning datasets as sources to construct a large-scale dataset, *Embodied-Points-200K*, which supports key embodied pointing capabilities. Then we train Embodied-R1 using a two-stage Reinforced Fine-tuning (RFT) curriculum with specialized multi-task reward design. Embodied-R1 achieves state-of-the-art performance on 11 embodied spatial and pointing benchmarks. Critically, it demonstrates robust zero-shot generalization by achieving a 56.2% success rate in the SIMPLEREnv and 87.5% across 8 real-world XArm tasks without any task-specific fine-tuning, representing a 62% improvement over strong baselines. Furthermore, the model exhibits high robustness against diverse visual disturbances. Our work shows that a pointing-centric representation, combined with an RFT training paradigm, offers an effective and generalizable pathway to closing the perception-action gap in robotics. More visualizations and datasets are available on website.

## 1 INTRODUCTION

Recent advancements in Vision-Language Models (VLMs) (Bai et al., 2025b; Team et al., 2025; Ni et al., 2025; 2024) have inspired a new wave of Vision-Language-Action (VLA) models (Kim et al., 2024) aimed at enhancing generalization in robotic manipulation. While these models exhibit strong visual perception and excel at imitating expert demonstrations, their manipulation performance degrades significantly in novel settings. This disparity is widely recognized as the *"seeing-to-doing gap"* (Yuan et al., 2025): a failure to reliably translate rich perceptual understanding into effective robotic actions. This gap is largely attributed to two key challenges: (a) *data scarcity*, where limited embodied data prevents from sufficiently grounding language and vision with physical actions (Walke et al., 2023; Lin et al., 2024), and (b) *heterogeneity*, where diverse robot morphologies pose a significant challenge to knowledge transfer.

To bridge this gap, we propose *pointing* as an intuitive and effective paradigm to connect high-level understanding with generalizable action. A point-centric representation (Yuan et al., 2024b; Deitke et al., 2024) unifies semantic and spatial information into a compact, embodiment-agnostic format. This approach is highly scalable, overcoming data scarcity by leveraging broad visual datasets—including real-world/synthetic robotic data, and internet data. Simultaneously, its embodiment-agnostic nature enables knowledge transfer across diverse robot platforms, resolving the heterogeneity challenge. This paradigm offers a promising path, avoiding the fundamental mismatch between physical actions and pre-trained data in end-to-end VLAs (Black et al., 2024) while mitigating the cascading errors of multi-model pipelines in modular methods (Liu et al., 2024a; Huang et al., 2024c). Despite a surge in related work, the pointing information from existing methods is often incomplete and too simplistic for complex decision-making, leading to limited generalization. They often offer only narrow forms of guidance, such as affordance point (Ji et al., 2025), object visual trace (Xu et al., 2025), or target

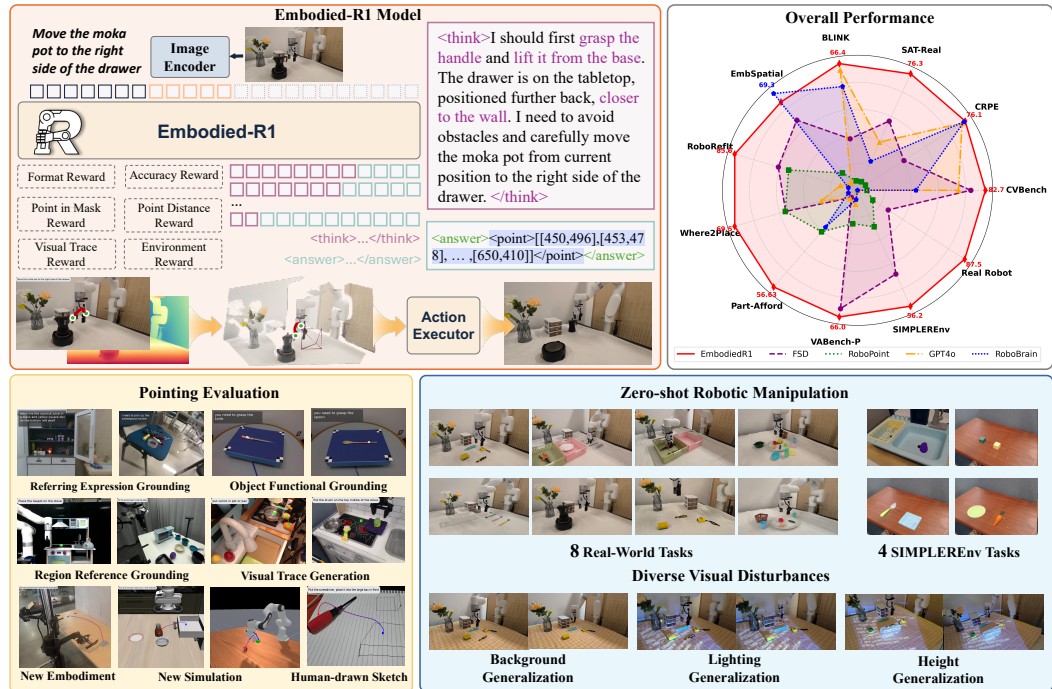

**Figure 1: Overview of the Embodied-R1 framework and its zero-shot manipulation performance.** Embodied-R1 performs explicit reasoning to generate "pointing" commands, enabling robust execution across a comprehensive suite of spatial reasoning benchmarks, pointing evaluations, and diverse real-world robotic tasks.

region (Yuan et al., 2024b). Crucially, even advanced models like FSD (Yuan et al., 2025), which anchor instructions through reasoning, are constrained by rigid CoT templates learned via Supervised Fine-Tuning (SFT). This rigid approach fundamentally limits their ability to generalize to novel tasks.

Therefore, we first systematize embodied pointing into four key capabilities in Fig. 2: Referring Expression Grounding (REG), Region Referring Grounding (RRG), Object Functional Grounding (OFG), and Visual Trace Generation (VTG). These abilities cover object identity ("What is this object?"), functional affordance ("How to use/grasp it?"), target location ("Where should it be placed?"), and can even implicitly convey the execution process ("How to complete the task?") through a visual trace. To develop these multi-task abilities, we constructed *Embodied-Points-200k*, a large-scale dataset of high-quality instances with verification methods, curated from diverse sources.

We introduce **Embodied-R1**, an advanced embodied reasoning VLM trained entirely with Reinforced Fine-tuning (RFT) to master manipulation through "pointing," as shown in Fig. 1. This training paradigm provides two critical advantages over standard SFT. First, it *enables flexible, free-form reasoning* beyond the rigid CoT templates, significantly enhancing generalization to novel tasks. Second, it *resolves the inherent multi-solution dilemma* in embodied pointing. For instance, an instruction to mark a point in an "empty space" has many valid solutions. While SFT struggles with this ambiguity and tends to overfit to a single data point, RFT can positively reinforce any correct answer, fostering a genuine understanding of the task's spatial constraints rather than mere memorization. With only 3B parameters, Embodied-R1 achieves state-of-the-art performance on multiple spatial reasoning and precise embodied pointing benchmarks. It achieves robust zero-shot manipulation by generating pointing signals as an intermediate representation, grounding its reasoning in the VLM's universal perception capabilities. This method preserves the model's inherent generalization by avoiding the prediction of low-level, embodiment-specific actions. Empirically, Embodied-R1 delivers state-of-the-art performance in the SIMPLEREnv (Li et al., 2024b) simulation, attains a remarkable 87.5% success rate in 8 real-world XArm tasks, and shows notably strong robustness against common visual disturbances such as changing light and backgrounds.

Our contributions include: ❶ pioneering "pointing" as a unified, embodiment-agnostic representation and defining core embodied pointing abilities to bridge perception and decision; ❷ constructing the comprehensive *Embodied-Points-200K* dataset for these capabilities; ❸ proposing Embodied-R1,

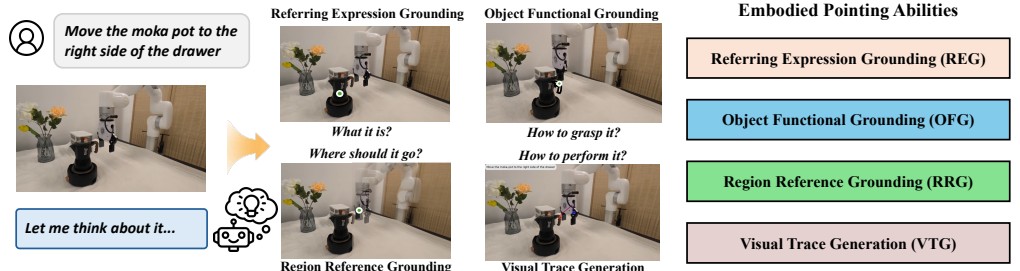

Figure 2: Overview of four embodied pointing abilities.

a VLM trained with RFT to resolve the multi-solution dilemma for embodied pointing, delivering powerful reasoning. ❹ With only 3B parameters, Embodied-R1 attains state-of-the-art performance on 11 diverse spatial and pointing benchmarks and enables robust zero-shot robotic manipulation, achieving 56.2% success in SIMPLEREnv simulation and 87.5% in 8 real-world XArm tasks, representing a 62% improvement over strong baselines, without task-specific fine-tuning.

## 2 RELATED WORK

**Embodied Reasoning in Robotic Manipulation** Enhancing the reasoning capabilities of Large Language Models (LLMs) is a pivotal research direction (Chu et al., 2025; OpenAI, 2024; Guo et al., 2025; Kimi-Team et al., 2025). In embodied AI, recent works have integrated reasoning into robotic manipulation, primarily through Supervised Fine-Tuning (SFT) with templated Chain-of-Thought (CoT) approaches. These works involve various forms of templated CoT, such as language prompts (Zawalski et al., 2024), visual subgoals (Zhao et al., 2025), or spatial relation graphs (Yuan et al., 2025) to guide execution. While newer efforts explore Reinforcement Fine-Tuning (RFT) (Lu et al., 2025; Liu et al., 2025a; Chen et al., 2025b) or latent planning (Huang et al., 2025), they are often limited to simulation or online learning. In contrast, our VLM stimulates free-form reasoning by integrating pointing with RL, avoiding fixed templates. Furthermore, Embodied-R1 utilizes pointing to precisely anchor reasoning within the scene and directly guide manipulation.

**Spatial Reasoning with VLMs** Spatial intelligence is essential for general-purpose embodied AI, enabling robots for precise manipulation (Yuan et al., 2024b; Song et al., 2024) and navigation (Hong et al., 2023; Li et al., 2024a). Current methods enhance the spatial reasoning of VLMs primarily through SFT on custom datasets (Du et al., 2024; Ray et al., 2024; Cheng et al., 2024; Chen et al., 2024). Some approaches also employ spatial CoT for step-by-step reasoning (Yuan et al., 2025; Liu et al., 2025b). Differently, Embodied-R1 employs RL to elicit emergent reasoning, which we show leads to stronger out-of-distribution (OOD) generalization compared to SFT-centric approaches.

**Visual Auxiliary Signals for Robotic Manipulation** Using visual auxiliary signals (Bharadhwaj et al., 2024; Wen et al., 2023; Xu et al., 2024; Zheng et al., 2024; Yuan et al., 2024a; Dong et al., 2024b) is a promising paradigm for abstracting away embodiment-specific details and enhancing cross-robot generalization. Prior works have explored various signals, including keypoints (Yuan et al., 2024b; 2025), affordance maps (Huang et al., 2024a; 2023; Li et al., 2024c; Wang et al., 2025a), bounding boxes (Liu et al., 2024a; Huang et al., 2024b), optical flow (Xu et al., 2024; Wen et al., 2023), and visual trajectories (Yuan et al., 2025; Ji et al., 2025; Li et al., 2025; Gu et al., 2023). In contrast to these methods, which typically generate a single type of visual signal, we propose a unified "pointing" definition to express diverse, multi-granular manipulation intents. We adopt an RL paradigm to explicitly improve zero-shot generalization in novel environments.

**Improving VLM Reasoning via RFT** Enhancing the reasoning capabilities of large models via RFT (Chu et al., 2025; Liu et al., 2024b) has emerged as a pivotal research direction, demonstrating superior performance in complex mathematics (Yu et al., 2025) and coding (Jimenez et al., 2024) tasks. This paradigm has been effectively extended to the multimodal domain, with numerous studies confirming that RFT significantly boosts multimodal perception and understanding capabilities (Tan et al.; Feng et al., 2025). Specifically, applications such as Seg-Zero (Liu et al., 2025c) introduce cognitive reasoning to guide segmentation, while VisionReasoner (Liu et al., 2025d) proposes a unified visual perception framework based on RFT. Building upon VLM foundation models, post-training

with curated data using algorithms like GRPO (Shao et al., 2024) or PPO (Schulman et al., 2017) can achieve performance and generalization capabilities that surpass standard SFT. This methodology has been successfully applied across diverse domains, including affordance prediction (Wang et al., 2025a), video understanding (Park et al., 2025). However, the application of RFT in embodied AI, particularly in robotic manipulation, remains underexplored. In this work, we propose an R1-style VLM tailored for robotic manipulation. By integrating a fine-grained and comprehensive "pointing" mechanism with RFT, we bridge the critical gap between "seeing" and "doing".

## 3 EMBODIED-R1: ADVANCING EMBODIED REASONING VIA RFT

This section first details the architecture and embodied pointing capabilities. We then describe dataset construction and training methodology, concluding with its deployment in real-world scenarios.

### 3.1 THE ARCHITECTURE AND CAPABILITIES OF EMBODIED-R1

Embodied-R1 is built upon the Qwen2.5-VL architecture (Bai et al., 2025a) and is specifically optimized for embodied manipulation by mastering four fundamental pointing abilities. These abilities all generate image coordinates $\mathbf{p} = (p, q) \in [0, w] \times [0, h]$, but differ in their semantic purpose and output structure. ❶ **Referring Expression Grounding (REG)** localizes an object from a linguistic description by generating a point within its segmentation mask. ❷ **Region Referring Grounding (RRG)** identifies a spatial region from relational language (e.g., "the space between objects") by generating a point in a suitable free-space location. Furthermore, ❸ **Object Functional Grounding (OFG)** identifies functionally critical parts of an object (i.e., affordances), such as a handle for grasping, by marking a point on that area. Finally, ❹ **Visual Trace Generation (VTG)** produces an ordered sequence of points, $\boldsymbol{\tau} = \{\mathbf{p}_t \mid t = 1, 2, \ldots, T\}$, $T$ denotes the sequence length, to form a complete, object-centric manipulation trajectory. This embodiment-agnostic path provides a comprehensive spatial plan for the robot to follow. Visualizations are presented in Fig. 2.

### 3.2 ENHANCING THE EMBODIED REASONING ABILITIES OF VLM

To develop general embodied pointing capabilities, Embodied-R1 is trained on three data types detailed in Fig. 3: embodied spatial reasoning for foundational awareness, general reasoning to preserve existing skills, and embodied pointing to learn the four key abilities.

**General and Spatial Reasoning Data** The foundation of reasoning data is **Embodied-Spatial-84K**, an embodied spatial awareness dataset aggregated from SAT (Ray et al., 2024) and WhatsUp (Kamath et al., 2023). For objective evaluation and verifiable rewards, all source data were converted into a unified multiple-choice format. Furthermore, to counteract the issue of catastrophic forgetting and preserve general reasoning during specialized training, we supplement this with **ViRL-subset-18K**, a diverse general-knowledge set. This 18K subset was curated from the ViRL (Wang et al., 2025b) dataset by filtering for difficulty and balancing content across subjects and types. This process yields the general-knowledge component of our composite dataset, creating a balanced curriculum that fosters specialized spatial skills while safeguarding the model's foundational knowledge.

**Embodied Pointing Data** To advance a suite of embodied pointing capabilities, we introduce the **Embodied-Points-200K** dataset, a high-quality, meticulously curated corpus containing about 200k samples. To address the multi-solution dilemma inherent in embodied pointing problems, we avoid constructing "question-answer" pairs typical for SFT. Instead, we structure the data as "question-verification" pairs, leveraging RFT for training. Subsequently, pre-defined reward functions for each task evaluate the response based on the verification and calculate the corresponding rewards. We briefly outline the pipeline for generating point data below. Data generation details are in App. A.

- **REG Data:** Precise localization is critical for robotic manipulation, but traditional bounding boxes suffer from inherent ambiguity. We therefore constructed a point-centric REG dataset, integrating web images from RefCOCO (Kazemzadeh et al., 2014) and embodied data from RoboRefIt (Lu et al., 2023; Yuan et al., 2024b) and RoboPoint (Yuan et al., 2024b) for broad coverage. We critically adjusted the success criterion: the model must output a single point instead of a bounding box. A prediction is considered correct if this point falls within the object's segmentation mask.

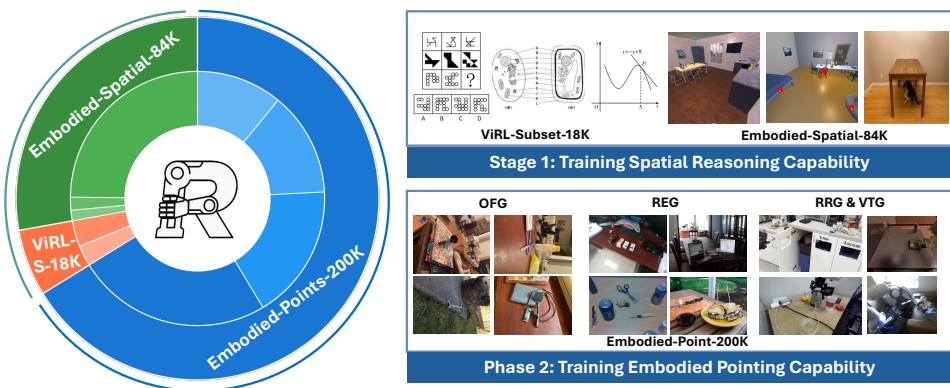

**Figure 3: Overview of training data**: In stage 1, we focus on improving the model's spatial reasoning capability, while incorporating a small amount of general reasoning data. In stage 2, we train the model's embodied pointing capabilities, which comprise four distinct capability items.

- **RRG Data:** To enable robots to comprehend complex spatial placement commands, we developed an automated data generation pipeline for creating relation-aware placement regions. This pipeline processes a large corpus of open-source embodied dataset (about 1 million), and after rigorous filtering, yields 33,000 high-quality samples. The core process includes: ❶ **Region Extraction:** extracting the final position of the manipulated object from the terminal frame; ❷ **Region Referring:** calculating the precise placement of the region relative to a reference object in the scene; and ❸ **Rendering:** rendering this spatial placement information back onto the initial image. To ensure data diversity and quality, we designed a heuristic filtering strategy, which covers a rich variety of spatial relationships, object configurations, and scenes.

- **OFG Data:** To cultivate Embodied-R1's fine-grained understanding of object affordances, we constructed a 40K-sample OFG dataset. We leveraged the HandAL dataset (Guo et al., 2023), converting its meticulous annotations of manipulable parts into bounding boxes for ground-truth verification. To promote generalization, we then employed GPT-4o to generate a diverse set of function-related questions (e.g., "Which part should be held when using a knife to cut vegetables?") corresponding to these functional parts.

- **VTG Data:** We constructed an object-centric visual trace dataset that exclusively tracks the object's movement. The extraction pipeline follows the methodology of (Yuan et al., 2025) and consists of three main steps: ❶ **Key Object Proposal:** Using GPT-4o to identify the primary object of interest for a given task. ❷ **KeyPoint Identification:** A self-supervised keypoint extractor (Huang et al., 2024c), in conjunction with Grounded-SAM (Ren et al., 2024), is used to automatically identify the object's grasping point. ❸ **Point Tracking and Projection:** We used Cotracker3 (Karaev et al., 2024) to compute the dense temporal visual trace originating from the keypoints. Next, the trajectory is then downsampled into 8 equidistant discrete points and projected back onto the initial image, creating an "image-visual trace" pair. Notably, using multiple pre-trained vision models in the process inevitably introduces noise. We implemented rigorous rule-based filtering and continually validated our approach using a manually annotated test set. Based on this feedback, we iteratively refined the filtering criteria to improve the quality of the dataset.

**Training Strategy** Based on the collected data, Embodied-R1 adopts a two-stage training process: the first stage focuses on enhancing spatial reasoning, as spatial reasoning serves as the foundation for point comprehension; the second stage trains embodied pointing capabilities using point-centric, multi-task mixed data. The optimization process is powered by the GRPO (Guo et al., 2025) algorithm. For any given input, the behavior policy generates multiple candidate responses, and their relative advantages are determined by normalizing rewards within the group. The model is then optimized using a clipped surrogate loss to maximize expected returns while maintaining training stability.

## 3.3 MULTI-TASK REWARD DESIGN

To stabilize multi-task training, where simpler tasks can dominate policy optimization, we design a modular reward system. Each task is assigned a total reward $\mathcal{R}$, which is a normalized weighted sum of components from a reward library. This scaling is crucial for balancing learning across tasks.

The following is the complete reward function library. ❶ **Format Rewards:** A binary reward $r_{\text{format}}(y) = \mathbb{I}(\texttt{tags valid}(y))$ enforces a structured output $y$, including required tags like `<think>` and a unified coordinate representation `<point>[[...]]</point>`. ❷ **Accuracy Rewards:** For general QA tasks, an accuracy reward $r_{\text{acc}}(y, g) = \mathbb{I}(y = g)$ is used, where $g$ is the ground-truth answer. ❸ **Point in Mask Reward:** For pointing tasks, point in mask reward function $r_{\text{mask}}$ is determined by whether the predicted output point $p$ lies within the ground-truth answer mask $M_{\text{gt}}$. The reward function can be formally expressed as $r_{\text{mask}}(\mathbf{p}, M_{\text{gt}}) = \mathbb{I}(\mathbf{p} \in M_{\text{gt}})$.

❹ **Point Distance Reward:** To improve learning efficiency, we also designed a dense auxiliary reward $r_{\text{dis}}$, which is used to guide the predicted point to approach the target region $M_{\text{gt}}$. The Euclidean distance is $d = \|\mathbf{p} - g\|_2$, where $g$ is the center of the $M_{\text{gt}}$. Given pixel distance thresholds $D_{\text{min\_thresh}}$ and $D_{\text{max\_thresh}}$, $r_{\text{dis}}$ is then defined as $r_{\text{dis}}(\mathbf{p}, M_{\text{gt}}) = \min\left(1.0, \max\left(0.0, 1.0 - \frac{d - D_{\text{min\_thresh}}}{D_{\text{max\_thresh}} - D_{\text{min\_thresh}}}\right)\right)$.

❺ **Visual Trace Reward:** For evaluating generated visual trace, rewards are derived from trajectory similarity metrics comparing the predicted trajectory $\tau$ with the ground-truth trajectory $\tau_{\text{gt}}$. First, we compare the number of points in the $\tau$ and $\tau_{\text{gt}}$. Using the longer one as the reference, we interpolate both trajectories to have the same number of points and then proceed with the calculation Root Mean Square Error (RMSE): $d_{\text{RMSE}}(\tau, \tau_{\text{gt}})$. Similarly, we use the $D_{\text{RMSE\_min}}$ and $D_{\text{RMSE\_max}}$ hyperparameters to ensure that the reward remains between 0 and 1. The reward is calculated as: $r_{\text{trace}}(\tau, \tau_{\text{gt}}) = \min\left(1.0, \max\left(0.0, 1.0 - \frac{d_{\text{RMSE}}(\tau, \tau_{\text{gt}}) - D_{\text{RMSE\_min}}}{D_{\text{RMSE\_max}} - D_{\text{RMSE\_min}}}\right)\right)$.

**Total Reward:** The total reward $\mathcal{R}$ for each task is formulated as a combination of these individual reward components. We define the reward function library $\mathcal{F} = \{r_{\text{format}}, r_{\text{acc}}, r_{\text{mask}}, r_{\text{dis}}, r_{\text{trace}}\}$. Each component function evaluates a specific aspect of the model's performance. Since we conduct mixed training on multiple tasks, in order to ensure that each task is equally and sufficiently trained, we constrain the total reward $\mathcal{R}$ for each task to the range of 0 to 1, which is implemented as follows. $\mathcal{R}$ is formulated as a weighted-sum combination: $\mathcal{R} = \sum_{r \in \mathcal{F}} w_r \cdot r$. The task-specific weights $w_r$ are normalized to sum to one ($\sum_{r \in \mathcal{F}} w_r = 1$). This structure guarantees that the total reward $\mathcal{R}$ is also bounded within the range $[0, 1]$ and allows us to tailor the reward signal for each task's specific needs. For example, the RRG task requires simultaneously satisfying the format requirements, ensuring that the predicted points are within the specified region, and accelerating training by employing dense distance rewards. $\mathcal{R}_{\text{RRG}}$ is defined as $\mathcal{R}_{\text{RRG}} = 0.1 r_{\text{format}} + 0.2 r_{\text{dis}} + 0.7 r_{\text{mask}}$. This consistent scaling of rewards across tasks is crucial for stabilizing. We refer to App. B for detailed hyperparameters.

## 3.4 ACTION EXECUTOR OF EMBODIED-R1

Through its pointing mechanism, Embodied-R1 can be flexibly integrated with various downstream action executors. This architecture allows Embodied-R1 to initiate reasoning at any task stage, dynamically select the required pointing ability, and couple its output with a motion planner for zero-shot robotic control. Besides, similar to Hamster (Li et al., 2025), Embodied-R1 can also serve as a high-level planner that integrates with learning-based methods to enable closed-loop control. Here, we implement this through two primary pipelines. **Affordance Points Branch (-P)**: In this branch, Embodied-R1 leverages its RRG and OFG capabilities to predict key points for grasping and placement. These points then serve as targets for a CuRobo (Sundaralingam et al., 2023) motion planner, which generates a collision-free trajectory for the robot's end-effector. **Visual Trace Branch (-V)**: This branch utilizes the object-centric visual traces generated by VTG. The 2D trace $\tau$ is first mapped to 3D Cartesian coordinates using the pinhole camera model and initial depth information. These discrete 3D points are then interpolated to form a continuous motion trajectory in SE(3) space, which the robot follows for execution, similar to the FSD (Yuan et al., 2025) methodology.

## 4 EXPERIMENTS

To validate Embodied-R1's generalization in robotic manipulation, we conducted extensive experiments evaluating its *Seeing* (spatial reasoning and pointing capabilities) and *Doing* (manipulation tasks) dimensions. Our evaluation encompassed 11 QA benchmarks, 4 simulated tasks (SIM-PLEREnv) (Li et al., 2024b), and 8 real-world robot (xArm platform) tasks. We used the Qwen2.5-VL-3B-Instruct (Bai et al., 2025a) model as the initial model. First, we trained using the Embodied-Spatial-84K and ViRLsubset-18K datasets. Then, we continued training with the second-stage

**Table 1:** Performance comparison on spatial reasoning benchmarks. **Bold** indicates the highest value among open-source models, and underlined values show the second-highest scores.

| | CVBench | | | | | CRPE | | | | SAT | | | BLINK | | | | EmbSp. | Rank |
|---|---|---|---|---|---|---|---|---|---|---|---|---|---|---|---|---|---|---|
| | Count | 2DRel | 3DDep | 3DDis | Avg. | Subj. | Pred. | Obj. | Avg. | Val | Real | MV | RelDepth | SpRel | Obj | Avg. | Test | |
| *Closed-source models* | | | | | | | | | | | | | | | | | | |
| GPT4V | 62.4 | 71.1 | 79.8 | 68.3 | 70.4 | 76.7 | 65.1 | 68.5 | 70.1 | 44.8 | 50.7 | 55.6 | 59.7 | 72.7 | 54.9 | 60.7 | 36.1 | - |
| GPT4o | 65.9 | 85.5 | 87.8 | 78.2 | 79.4 | 81.9 | 71.8 | 73.6 | 75.8 | 49.4 | 57.5 | 60.2 | 74.2 | 69.2 | 59.8 | 65.9 | 49.1 | - |
| *Open-source models* | | | | | | | | | | | | | | | | | | |
| LLaVA-1.5-13B | 58.2 | 46.6 | 53.0 | 47.8 | 51.4 | 57.4 | 54.2 | 55.2 | 55.6 | 51.4 | 41.6 | 41.4 | 53.2 | 69.9 | 52.5 | 54.2 | 35.1 | 9.4 |
| SAT-Dynamic-13B | 61.5 | 89.7 | 80.7 | 73.0 | 76.2 | 60.6 | 57.6 | 65.2 | 61.1 | **87.7** | 54.9 | 44.4 | 73.4 | 66.4 | 45.9 | 57.5 | 51.3 | 6.6 |
| RoboPoint-13B | 56.5 | 77.2 | 81.5 | 57.7 | 68.2 | 66.3 | 62.4 | 70.9 | 66.5 | 53.3 | 46.6 | 44.4 | 62.1 | 65.7 | 56.6 | 57.2 | 51.4 | 7.2 |
| ASMv2-13B | 58.9 | 68.9 | 68.9 | 68.9 | 66.4 | 69.2 | 59.0 | 65.3 | 64.5 | 63.9 | 46.7 | 44.4 | 56.5 | 65.0 | **63.9** | 57.5 | 57.4 | 7.2 |
| FSD-13B | 62.4 | 86.5 | **88.0** | **86.7** | 80.9 | 75.2 | 65.1 | 70.4 | 70.2 | 73.2 | 63.3 | 46.6 | 70.2 | 78.3 | 46.7 | 60.5 | 63.3 | 4.6 |
| RoboBrain-7B | 64.3 | 76.6 | 84.0 | 72.0 | 74.2 | 81.3 | **71.8** | 74.8 | 76.0 | 45.3 | 52.2 | **55.6** | 75.8 | **81.8** | 45.1 | 64.6 | **69.3** | 4.4 |
| Qwen2.5VL-3B | 68.4 | 72.8 | 77.0 | 68.2 | 71.6 | 80.7 | 71.0 | **76.1** | 76.0 | 48.7 | 45.1 | 44.4 | 66.9 | 79.7 | 55.7 | 61.7 | 62.8 | 5.6 |
| Embodied-SFT | 66.4 | **92.3** | 85.8 | 83.8 | 82.1 | 74.7 | 71.3 | 73.8 | 73.3 | 59.3 | 65.5 | 50.4 | **81.5** | 78.3 | 54.9 | 66.3 | 63.1 | 3.7 |
| Embodied-R1 w/o CS | 70.4 | 90.2 | 84.5 | 81.0 | 81.5 | 80.3 | 69.9 | 75.4 | 75.2 | 70.0 | 73.9 | 47.4 | 72.6 | 79.7 | 56.6 | 64.1 | 65.4 | 3.4 |
| Embodied-R1 | **70.6** | 90.8 | 84.7 | 84.8 | **82.7** | **82.2** | 70.7 | 75.4 | **76.1** | 70.0 | **76.3** | 51.1 | 76.6 | 80.4 | 57.4 | **66.4** | 67.4 | **2.1** |

**Table 2: Performance on 4 Pointing benchmarks.** The score is the accuracy of points falling within the target region.

| Model | RoboRefit | Where2Place | VABench-P | Part-Afford |
|---|---|---|---|---|
| GPT4o | 15.28 | 29.06 | 9.30 | 10.15 |
| ASMv2 | 48.40 | 22.00 | 10.07 | 13.75 |
| RoboBrain | 10.10 | 16.60 | 7.00 | 25.25 |
| RoboPoint | 49.82 | 46.01 | 19.09 | 27.60 |
| FSD | 56.73 | 45.81 | 61.82 | 9.55 |
| Qwen2.5VL | 74.90 | 31.11 | 9.89 | 23.42 |
| Embodied-SFT | 83.85 | 41.25 | 50.46 | 40.20 |
| Embodied-R1 | **85.58** | **69.50** | **66.00** | **56.63** |

**Table 3: Performance on VABench-V**. Lower values are better for RMSE/MAE, higher is better for LLM Score.

| Model | RMSE ↓ | MAE ↓ | LLM Score ↑ |
|---|---|---|---|
| GPT-4o | 136.1 | 113.5 | 4.4 |
| DINOv2 Predictor | 128.3 | 117.5 | 4.0 |
| RoboBrain | 121.6 | 103.8 | 4.5 |
| FSD | 78.3 | 63.4 | 6.2 |
| Embodied-SFT | 109.4 | 65.2 | 6.2 |
| Embodied-R1 | **77.8** | **45.0** | **7.3** |

EmbodiedPoints-200K dataset. For all experiments, we focus on comparing SFT models trained with the same batch size and data, which we refer to as Embodied-SFT. For training details, please refer to App. B and App. C. We also provide additional experiments in App. F.

## 4.1 EVALUATION OF SPATIAL REASONING CAPABILITIES

**Setup:** To evaluate the foundational spatial reasoning from our first training stage, we benchmarked Embodied-R1 on five diverse benchmarks: CVBench (Tong et al., 2024), BLINK (Fu et al., 2024), CRPE (Wang et al., 2025c), SAT (Ray et al., 2024), and EmbSpatial-Bench (Du et al., 2024). Baselines included leading closed-source models (GPT-4o, GPT-4V) and various open-source, spatially-enhanced VLMs such as SAT-Dynamic (Ray et al., 2024), RoboPoint (Yuan et al., 2024b), and FSD (Yuan et al., 2025). We also included two key ablations: Embodied-R1 w/o CS, which excludes the ViRL common-sense dataset, and Embodied-SFT, a variant trained only with SFT.

**Results:** As shown in Tab. 1, Embodied-R1 demonstrates state-of-the-art performance among open-source models **with only 3B parameter**. It achieves an average rank of 2.1, significantly outperforming its variants trained without common-sense data (Embodied-R1 w/o CS, Rank 3.4) or with only SFT (Embodied-SFT, Rank 3.7). We believe that more diverse data can stimulate exploratory reasoning capabilities. Embodied-R1 surpasses larger, specialized embodied models, including RoboBrain-7B and FSD-13B, underscoring the effectiveness of our RFT training strategy.

## 4.2 EVALUATION OF POINTING CAPABILITIES

**Setup:** To comprehensively evaluate Embodied-R1's embodied pointing abilities, we tested it across our four defined capabilities. For REG, we used RoboRefIt (Lu et al., 2023), which challenges models with relational references between similar objects. For RRG, we selected Where2Place (Yuan et al., 2024b) and the more complex, reasoning-intensive VABench-P (Yuan et al., 2025). We assessed OFG on our custom Part-Afford Benchmark, derived from RGBD-Part-Affordance (Myers et al., 2015) by filtering 2000 grasp-related affordances. This benchmark encompasses 105 types of kitchen, workshop, and gardening tools, designed to evaluate the generalization capability of affordance prediction in OOD scenarios. Finally, we followed the VABench-V (Yuan et al., 2025) evaluation methodology for VTG capacity, measuring MAE, RMSE, and LLM Scores.

**Results** are presented in Tabs. 2 and 3, with visualizations in Fig. 4. Key observations are as follows:

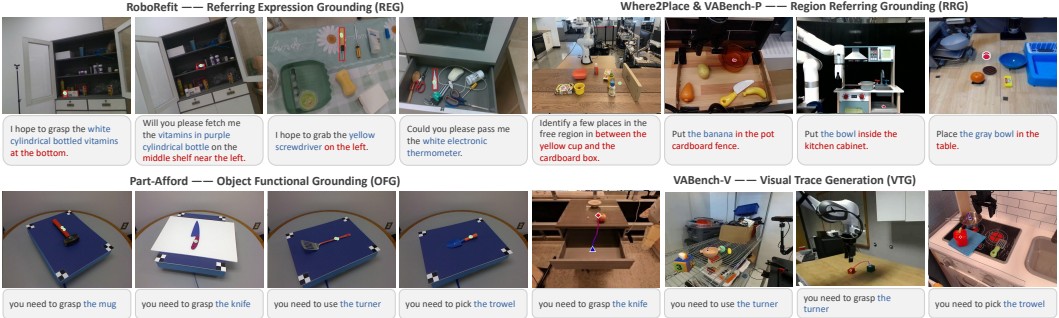

**Figure 4: Visualizing Embodied-R1's Performance on Various Pointing Tasks.** The model can follow diverse text instructions and generalize its capabilities to novel, unseen environments.

(*O1*) **Powerful general VLMs perform poorly on pointing tasks**, such as GPT-4o and Qwen2.5-VL, indicating the necessity of specialized training with data for these capabilities. This indicates that it is necessary to train embodied VLMs with strong spatial reasoning and pointing abilities.

(*O2*) **Embodied-R1 demonstrates superior performance across key benchmarks**. In Tabs. 2 and 3, across all benchmarks for REG, RRG, OFG, and VTG, Embodied-R1 consistently outperforms both general and specialized baselines, including other pointing-focused models like FSD and RoboPoint. As shown in Fig. 4, a single Embodied-R1 model masters this diverse skill set, demonstrating high accuracy even with small objects and complex spatial relationships in cluttered scenes.

(*O3*) **Embodied-R1 generates highly accurate visual traces for robotic manipulation.** On the VABench-V benchmark, Embodied-R1 achieves the lowest RMSE and MAE, indicating its ability to produce precise point sequences for traces, a crucial aspect for reliable action execution. Embodied-R1 also demonstrates significant improvement, indicating that R1 focuses on reasoning about the ideal visual trajectory rather than rote memorization. We refer to more visualization in App. E.

(*O4*) **Embodied-R1 significantly outperforms models trained solely with SFT**. Compared to the Embodied-SFT, Embodied-R1 demonstrates substantial improvements across these tasks, validating the benefits of RFT in developing strong generalization capabilities for embodied pointing.

**Table 4: SimplerEnv Evaluation on WidowX Robot.** Each task is tested 24 episodes. Most of the results for end-to-end VLAs are sourced from Chen et al. (2025a), while the results for the remaining models are reproduced in accordance with the official code.

| Type | Model | Put Spoon on Towel | Put Carrot on Plate | Stack Green Block on Yellow Block | Put Eggplant in Yellow Basket | Avg |
|---|---|---|---|---|---|---|
| *End-to-end VLAs* | Octo (Team et al., 2024) | 41.7 | 8.2 | 0.0 | 56.7 | 26.7 |
| | $\pi_0$ (Black et al., 2024) | 29.1 | 0.0 | 16.6 | 62.5 | 27.1 |
| | $\pi_0$-fast (Pertsch et al., 2025) | 29.1 | 21.9 | 10.8 | 66.6 | 48.3 |
| | OpenVLA (Kim et al., 2024) | 4.2 | 0.0 | 0.0 | 16.7 | 5.2 |
| | OpenVLA-OFT (Kim et al., 2025) | 34.2 | 30.0 | 30.0 | **72.5** | 41.8 |
| | ThinkAct (Huang et al., 2025) | 58.3 | 37.5 | 8.7 | 70.8 | 43.8 |
| | Magma (Yang et al., 2025) | 37.5 | 31.0 | 12.7 | 60.5 | 35.4 |
| *Modular Methods* | MOKA (Liu et al., 2024a) | 45.8 | 41.6 | 33.3 | 12.5 | 33.3 |
| | Sofar (Qi et al., 2025) | 55.5 | 56.9 | **62.5** | 40.2 | 53.8 |
| *Affordance Methods* | RoboPoint (Yuan et al., 2024b) | 16.7 | 20.8 | 8.3 | 25.0 | 17.7 |
| | FSD (Yuan et al., 2025) | 41.6 | 50.0 | 33.3 | 37.5 | 40.6 |
| | Embodied-R1 | **62.5** | **68.0** | 36.1 | 58.3 | **56.2** |

## 4.3 EVALUATION OF EMBODIED-R1 FOR ROBOT MANIPULATION

**SimplerEnv Simulation.** To rigorously assess the zero-shot generalization capability of Embodied-R1, we conducted experiments on the WidowX arm using SimplerEnv (Li et al., 2024b), employing a CuRobo (Sundaralingam et al., 2023) planner for execution. We compared performance against a comprehensive suite of baselines across three categories: (1) **End-to-end VLAs**, including standard models (Octo, OpenVLA, $\pi_0$) and stronger variants ($\pi_0$-fast, OpenVLA-OFT, Magma, ThinkAct); (2) **Modular Methods** (MOKA, Sofar); and (3) **Affordance Methods** (RoboPoint, FSD). As presented

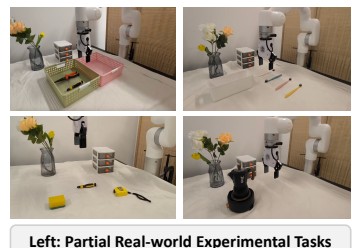 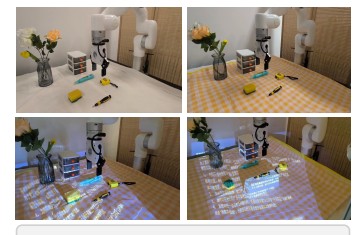 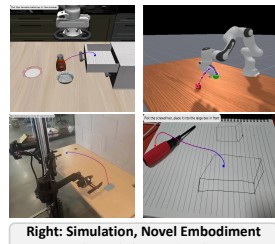

**Left: Partial Real-world Experimental Tasks** | **Mid: Visual Disturbances Tasks** | **Right: Simulation, Novel Embodiment and Hand-drawn Sketches**

Figure 5: **Left:** Snapshots from a variety of our real-world experimental manipulation tasks. **Middle:** Demonstration of the robustness under significant visual disturbances, such as background and lighting changes. **Right:** Specifically designed tests evaluating the zero-shot generalization of VTG in entirely OOD scenarios, including simulation, a novel robotic embodiment, and hand-drawn sketches. See App. E for complete execution process.

Table 5: **Real-world experimental evaluation results**. The first two tasks were conducted 5 times each, while the other tasks were conducted 6 times each. The best results are highlighted in bold. **[x]:** The instruction for each trial is a randomly selected color. **Nearest object\***: the object closest to the camera's viewpoint.

| | Pick up the strawberry | Move the egg to the bowl | Move the vise to the red basket | | Place the fork in the green bin | | Pick the [x] toothbrush and place it to the bucket | | Move the nearest object* to the right side of the drawer | | Put the screwdriver between drawer and the vase | | Move the moka pot to the right of drawer | | Avg |
|---|---|---|---|---|---|---|---|---|---|---|---|---|---|---|---|
| | Succ. | Succ. | Grasp. | Succ. | Grasp. | Succ. | Correct Obj. | Succ. | Correct Obj. | Succ. | Grasp. | Succ. | Grasp. | Succ. | Succ. |
| MOKA | 0.0% | 40.0% | 0.0% | 0.0% | 16.7% | 0.0% | 16.7% | 16.7% | 83.3% | 16.7% | 83.3% | 0.00% | 16.7% | 0.00% | 9.2% |
| RoboPoint | 40.0% | 60.0% | 50.0% | 0.0% | 0.0% | 0.0% | 16.7% | 0.0% | 0.0% | 0.0% | 66.7% | 0.0% | 0.0% | 0.0% | 12.5% |
| FSD | 20.0% | 80.0% | 66.7% | 33.3% | 16.7% | 16.7% | 16.7% | 0.0% | 0.0% | 0.0% | 66.7% | 33.3% | 16.7% | 16.7% | 25.0% |
| **Embodied-R1-P** | **100.0%** | **100.0%** | 66.7% | 50.0% | **100.0%** | **100.0%** | **100.0%** | **83.3%** | **100.0%** | **100.0%** | **100.0%** | **100.0%** | 50.0% | **33.3%** | 83.3% |
| **Embodied-R1-T** | **100.0%** | **100.0%** | **100.0%** | **100.0%** | **100.0%** | **100.0%** | **100.0%** | 66.7% | **100.0%** | **100.0%** | **100.0%** | **100.0%** | 33.3% | **33.3%** | **87.5%** |

Table 6: **Performance Comparison between SFT and RL on RRG benchmarks.**

| RL | Think | Where2Place | VABench-P |
|---|---|---|---|
| ✓ | ✓ | **65.50** | **65.39** |
| ✓ | ✗ | 63.00 | 60.50 |
| ✗ | ✓ | 41.25 | 47.67 |
| ✗ | ✗ | 36.85 | 50.46 |

Table 7: **Performance of Embodied-R1 under visual disturbances.** BC: Background Change, LC: Light Change, HC: Height Change.

| Disturbance | Grasp (%) | Succ. (%) |
|---|---|---|
| Original | 100 | 100 |
| BC | 100 | 100 |
| BC+LC | 83 | 83 |
| BC+LC+HC | 83 | 83 |

in Tab. 4, Embodied-R1 achieves a state-of-the-art average success rate of 56.2%. Notably, while recent end-to-end methods such as $\pi_0$-fast (48.3%) and OpenVLA-OFT (41.8%) show improved robustness compared to the base OpenVLA (5.2%), Embodied-R1 still outperforms them significantly. Furthermore, Embodied-R1 surpasses the strongest modular baseline (Sofar, 53.8%) and affordance baseline (FSD, 40.6%). These results suggest that explicit visual reasoning provides superior zero-shot generalization compared to end-to-end policy learning, particularly when facing unseen instructions and background variations without domain-specific fine-tuning.

**Real-World Robot Evaluation.** We conducted zero-shot real-world evaluations on an xArm 6 robot across eight tabletop manipulation tasks. The setup used a third-person Intel RealSense L515 camera (640×480), with all objects, scenes, and tasks being OOD to test generalization. As shown in Tab. 5, Embodied-R1 achieves an 87.5% zero-shot success rate, an improvement of over 60% compared to the RoboPoint and FSD baselines. We attribute this significant improvement to the baselines' poor performance on tasks requiring spatial reasoning (e.g., moving the nearest object) and their low success rates in grasping challenging rigid objects like a screwdriver. In contrast, the Embodied-R1 correctly identified these cases and achieved high success rates, demonstrating the effectiveness of reasoning process. Overall, Embodied-R1-V generated more accurate annotations than Embodied-R1-P, resulting in a slightly higher average success rate. In App. D, we conducted an in-depth analysis of failure cases and execution time.

To further test robustness, we selected a task requiring spatial reasoning and introduced zero-shot visual disturbances, including changes in background, lighting, and height. As shown in Tab. 7, Embodied-R1 demonstrated outstanding generalization against all disturbances. Surprisingly, it located the target and completed the task even under the poorest lighting conditions, while background changes had no impact on performance. This experiment validates that pointing serves as a universal representation that maintains both performance and robustness under visual distractions.

**Qualitative Analysis in Complex Scenarios.** To further validate the versatility of Embodied-R1, we conducted qualitative experiments across long-horizon, reasoning-intensive, and contact-rich

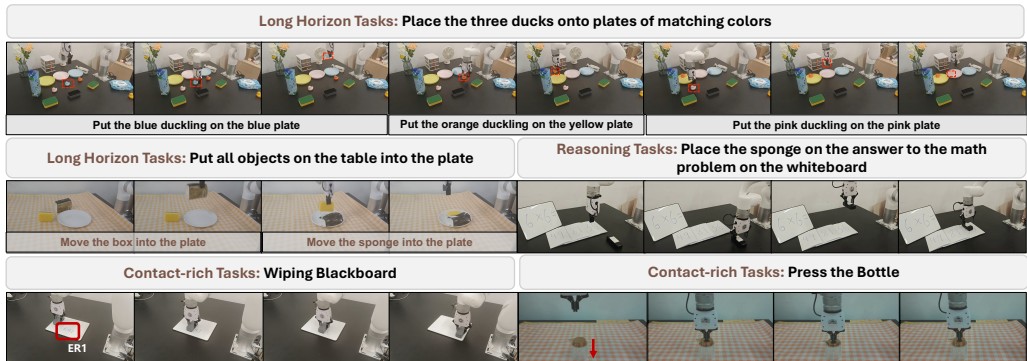

**Figure 6: Qualitative Analysis on Long-horizon, Reasoning and Contact-Rich Tasks.**

tasks (Fig. 6). (1) **Long-Horizon Tasks:** For complex scenarios such as *table organization* and *color-matched duck sorting*, we employed Gemini-2.5-Pro as a high-level planner to decompose global instructions into sub-tasks. Embodied-R1 then generated precise visual traces for each stage, successfully managing multi-step sequences. (2) **Reasoning-Driven Tasks:** In tasks requiring logical deduction (e.g., *"Place sponge on the answer to the math problem"*), the model demonstrated robust spatio-logical reasoning. It successfully localized semantic targets derived from internal calculations, effectively bridging the gap between abstract reasoning and spatial grounding. (3) **Contact-Rich Manipulation:** Embodied-R1 serves as a flexible interface that decouples high-level perception from low-level dynamics. In *blackboard wiping*, Embodied-R1's visual traces were executed via an impedance controller to maintain consistent contact force. For *bottle pressing*, we integrated visual localization with a force-aware Diffusion Policy (Chi et al., 2023) trained on 50 demonstrations. This synergy allowed the system to adapt to varying bottle positions while maintaining smooth, force-sensitive execution. These results indicate that while pointing serves as a geometric abstraction, it functions as a pivotal bridge linking high-level semantic understanding with diverse low-level control strategies, enabling robust generalization in complex embodied tasks.

## 4.4 FURTHER ANALYSIS AND ABLATIONS

**Embodied-R1 Exhibits Strong Generalization.** Despite being trained exclusively on real-world data, Embodied-R1 demonstrates remarkable zero-shot generalization on VTG tasks across entirely unseen scenarios (Fig. 5 (right)). It makes accurate predictions in simulations, suggesting a promising direction for sim-to-real transfer. Furthermore, Embodied-R1 adapts to different embodiments, highlighting the advantage of our embodiment-agnostic visual traces. Notably, its accuracy holds even in abstract scenarios, correctly marking a real screwdriver on a hand-drawn box.

**Ablation on SFT vs. RL.** To dissect our training paradigm, we compared RL against SFT and analyzed the impact of an explicit reasoning process. We trained four variants on RRG benchmarks. The results in Tab. 6 show that RL-based models consistently outperform their SFT counterparts, highlighting the crucial role of RL in OOD generalization. Our full model (RL w/ Think) performed the best, confirming that the reward-driven paradigm is most effective when combined with reasoning. In App. F, we provide further experiments, including an analysis of integration with learning-based methods, an ablation study on the benefits of mixed training, and results using RGB-D inputs.

## 5 CONCLUSION

We introduce Embodied-R1, an embodied reasoning VLM that bridges the critical "seeing-to-doing" gap in robotic manipulation. By training Embodied-R1 with a two-stage RFT paradigm on our large-scale curated dataset, we significantly enhance its spatial reasoning and embodied pointing abilities. Through its core pointing mechanism, Embodied-R1 masters a suite of capabilities, including grounding, spatial referencing, affordance marking, and visual trace generation, which are further applied to downstream robotic manipulation. Empirically, Embodied-R1 achieves state-of-the-art results across multiple benchmark tests and demonstrates robust zero-shot generalization in robotic manipulation tasks, offering a promising pathway toward more capable and general-purpose embodied AI. A detailed discussion of limitations is provided in App. G.

## ACKNOWLEDGEMENT

This work is supported by the National Natural Science Foundation of China Youth Student Basic Research Project (Grant No. 625B2128). This work is also supported by the National Natural Science Foundation of China (Grant Nos. 62422605, 62533021, 92370132) and the National Key Research and Development Program of China (Grant No. 2024YFE0210900). Yifu Yuan is also supported by the Young Science and Technology Scientists Sponsorship Program by CAST - Doctoral Student Special Plan. Pengyi Li is supported by the Basic Research Project (Grant No. 624B2101). We would like to thank Zhongwen Xu, Liang Wang, Shuyang Gu, and Chen Li for their participation in the discussions of this paper and for providing valuable insights. In addition, we would especially like to thank Yiyang Huang for the constructive suggestions on improving the figures in the manuscript.

## REPRODUCIBILITY STATEMENT

To promote transparency and reproducibility within the scientific community, we provide detailed training parameters and resources in the appendix. The complete code for both training and inference has been uploaded to the GitHub repository: https://github.com/pickxiguapi/Embodied-R1. In addition, we provide video demos of model execution and more visualization examples on the project homepage; we refer readers to https://embodied-r1.github.io/ for more information. We have made datasets and model checkpoints publicly available.

## ETHICS STATEMENT

This paper is dedicated to advancing the field of robotic manipulation towards the creation of more effective and versatile robotic assistants. Our research strictly adheres to responsible practices and aligns with the ICLR Code of Ethics. All training data was sourced from large-scale, open-access robotics datasets, with all assets utilized in full compliance with their original licensing and terms of service. We recognize that while the intended applications of this research are positive, the long-term societal impacts of increasing robotic autonomy warrant careful consideration.

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

# A   AUTOMATIC DATA GENERATION PIPELINE

In this section, we provide additional explanations regarding the generation of certain datasets. The generation processes of both the RRG and VTG datasets are improved based on Yuan et al. (2025).

**3D RRG Data Generation using Isaac Gym Simulation** Furthermore, we leveraged the Isaac Gym simulation engine to generate a synthetic dataset of 3D object rearrangements, equipping the model with 3D spatial awareness. In this task, which takes RGB-D images as input, the model is required to place objects in the correct relative positions according to instructions. Task success is automatically determined and fed back by the simulation based on the true physical state. The dataset comprises 10,028 tasks, each situated in a tabletop scene containing multiple objects. The input for each task consists of processed RGB and depth images, accompanied by a language instruction describing the desired target position of an object within the scene (e.g., "place the cup between the book and the spoon"). Based on these instructions, the model is required to output the 3D position of the target object in the camera coordinate system, specified by its pixel coordinates $(X, Y)$ and a depth value $D$ in millimeters. The $D$ value is obtained either through monocular depth estimation or by reasoning from the scene geometry. The generation process of our dataset is informed by the methodology of Open6Dor Qi et al. (2025). The object set utilized contains over 200 items spanning more than 70 distinct categories, originally sourced from the YCB Xiang et al. (2017) and Objaverse-XL Deitke et al. (2023) datasets. These objects underwent a rigorous selection process to ensure their physical integrity and semantic suitability for tabletop arrangements. All selected objects were subsequently scale-normalized and uniformly represented using a consistent mesh format.

In terms of scene configuration, between two and five objects were randomly selected from the object set and placed on a tabletop with random initial poses. For each configured scene, we rendered both RGB and depth images. The depth values represent ground truth measurements, with the scene's depth range spanning from 600 mm to 1700 mm. For subsequent processing convenience, the depth images were normalized to an 8-bit grayscale format (0-255). We filtered out low-quality scenes, such as those exhibiting implausible object placements or severe occlusions. To augment the dataset's quality and volume, we expanded a subset of high-quality, filtered data by algorithmically generating variations in task descriptions, such as substituting directional prepositions or altering object relations. Then, the task instructions are formulated in two primary types: basic directional commands (e.g., left, right, top, behind, front) and relational commands (e.g., "between," "center of"). All instructions adhere to a standardized template, for instance, "Place object A in front of object B," where A and B are objects present in the scene. During training, the model receives RGB and depth image inputs and is required to output the target's coordinates $(X, Y)$ and depth value $D$. The simulated environment then executes and evaluates the predicted position, giving positive rewards for correct predictions.

**VTG Dataset Generation Pipeline** For each video sequence, we first process the initial frame to perform instance segmentation on the manipulated object, thereby obtaining its pixel-wise mask. Instead of relying on a single tracking point, which is susceptible to tracking failure from occlusion or rapid motion, we sample a set of three distinct query points from within this mask. This multi-point initialization serves as a redundancy measure, significantly enhancing the robustness of the tracking process. These points are strategically chosen to represent the object's initial state before they are passed, along with the full video sequence, to the tracking model for trajectory prediction. The core of the trajectory generation is handled by the CoTracker model Karaev et al. (2024), which takes the initialized query points and video as input. The model concurrently tracks each point throughout the sequence, yielding a set of three candidate trajectories. As these trajectories may vary in quality and completeness due to transient tracking errors, a selection criterion is required to identify the single most representative path. We employ a path length heuristic for this purpose, calculating the total Euclidean distance of each trajectory. The trajectory with the longest path is selected as the definitive motion path for the object. The rationale behind this criterion is that the longest trajectory is most likely to have successfully tracked the object through its entire course of motion without premature termination. Following the selection of the representative trajectory, a two-stage refinement process is applied to produce the final visual trace. The raw trajectory, composed of discrete frame-by-frame coordinates, is first smoothed using cubic spline interpolation. This step transforms the discrete points into a continuous, smooth curve, effectively filtering out high-frequency noise and jitter inherent in the tracking process. From this smoothed curve, we then uniformly sample eight equidistant points. This final set of eight points constitutes the visual trace—a structured, discretized representation of the object's motion, suitable for downstream analysis and model consumption.

We found that this process inevitably faces prediction errors from pre-trained visual models, such as incorrect object grounding or incomplete motion trajectory tracking. To mitigate these issues, we employ stringent rule-based filtering methods using hyperparameters such as size thresholds and trajectory length thresholds. Before annotating each dataset, we iteratively adjust these rules and conduct manual sampling inspections. Only when the filtering rules are robust enough do we apply them to the full data generation pipeline, ensuring the high quality of the data.

## B  IMPLEMENTATION DETAILS OF EMBODIED-R1

**Training Hyperparameters:** We conducted model training on eight NVIDIA A100 40G GPUs. The first phase was trained for 2 epochs, and the second phase for 1 epoch, with each phase taking approximately 48 hours. The backbone model used is Qwen2.5-VL-3B-Instruct [1], with a maximum context length of 4096 and a maximum response length of 2048. The optimizer selected is AdamW, with a learning rate of 1e-6 and a weight decay coefficient of 1e-2. In Embodied-R1, we performed reinforcement learning training based on GRPO Shao et al. (2024), set the number of samples to 8, and introduced a KL penalty (coefficient 1e-2), with a global batch size of 128 for each step. For all experiments, we focus on comparing SFT models trained with the same batch size and data, which we refer to as Embodied-SFT. As for Embodied-SFT, we used exactly the same data but trained with a supervised learning loss, kept the batch size at 128, and trained for 3 epochs. The Embodied-R1 model is trained entirely based on the EasyR1 open-source codebase [2].

**Reward Hyperparameters:** To enable stable multi-task training, we constrain the total reward for each task to the range of 0 to 1 and define the total reward $\mathcal{R}$ as a weighted combination $\mathcal{R} = \sum_{r \in \mathcal{F}} w_r \cdot r$. Each task utilizes a different combination of reward terms. Here, we provide the hyperparameters for each task in the Tab. 8. We would like to add two clarifying points: First, if the task output fails to meet the required parsing format, subsequent analysis cannot proceed successfully, so the reward is set directly to 0. Second, for the VTG task, we introduced an additional constraint on the format: the generated visual trace must consist of *exactly 8 points*. In practice, we found that without this constraint, the model in the VTG task was prone to reward hacking behavior. It would tend to output only two points to form a straight line, which easily yields a high reward and prematurely terminates exploration. By enforcing the 8-point constraint, we compel the model to perform more complex curve interpolation, thereby improving the performance of visual trace generation. We provide a detailed comparison in the App. F.

**Table 8:** Detailed Reward Functions for Each Task

| Task | Reward Function $\mathcal{R}$ |
|---|---|
| General QA | $\mathcal{R} = 0.1\,r_{\text{format}} + 0.9\,r_{\text{acc}}$ |
| Spatial QA | $\mathcal{R} = 0.1\,r_{\text{format}} + 0.9\,r_{\text{acc}}$ |
| REG | $\mathcal{R} = 0.1\,r_{\text{format}} + 0.9\,r_{\text{mask}}$ |
| RRG | $\mathcal{R} = 0.1\,r_{\text{format}} + 0.6\,r_{\text{mask}} + 0.3\,r_{\text{dis}}$ |
| 3D RRG | $\mathcal{R} = 0.1\,r_{\text{format}} + 0.9\,r_{\text{env}}$ |
| OAG | $\mathcal{R} = 0.1\,r_{\text{format}} + 0.8\,r_{\text{mask}} + 0.1\,r_{\text{dis}}$ |
| VTG | $\mathcal{R} = 0.1\,r_{\text{format}} + 0.9\,r_{\text{trace}}$ |

---

[1] https://huggingface.co/Qwen/Qwen2.5-VL-3B-Instruct
[2] https://github.com/hiyouga/EasyR1

## C Embodied-R1 Prompts for Each Task

---

**Referring Expression Grounding (REG) Prompt**

Provide one or more points coordinate of object region this sentence describes: *Your Instruction*. The results are presented in a format <point>[[x1,y1], [x2,y2], ...]</point>. You FIRST think about the reasoning process as an internal monologue and then provide the final answer. The reasoning process and answer are enclosed within <think> </think> and <answer> </answer> tags. The answer consists only of several coordinate points, with the overall format being: <think> reasoning process here </think><answer><point>[[x1, y1], [x2, y2], ...]</point></answer>

---

**Region Referring Grounding (RRG) Prompt**

You are currently a robot performing robotic manipulation tasks. The task instruction is: *Your Instruction*. Use 2D points to mark the target location where the object you need to manipulate in the task should ultimately be moved. You FIRST think about the reasoning process as an internal monologue and then provide the final answer. The reasoning process and answer are enclosed within <think> </think> and <answer> </answer> tags. The answer consists only of several coordinate points, with the overall format being: <think> reasoning process here </think><answer><point>[[x1, y1], [x2, y2], ...]</point></answer>.

---

**Object Functional Grounding (OFG) Prompt**

Please provide the 2D points coordinates of the region this sentence describes: *Your Instruction*. The results are presented in a format <point>[[x1,y1], [x2,y2], ...]</point>. You FIRST think about the reasoning process as an internal monologue and then provide the final answer. The reasoning process and answer are enclosed within <think> </think> and <answer> </answer> tags. The answer consists only of several coordinate points, with the overall format being: <think> reasoning process here </think><answer><point>[[x1, y1], [x2, y2], ...]</point></answer>.

---

**Visual Trace Generation (VTG) Prompt**

You are currently a robot performing robotic manipulation tasks. The task instruction is: *Your Instruction*. Use 2D points to mark the manipulated object-centric waypoints to guide the robot to successfully complete the task. You must provide the points in the order of the trajectory, and the number of points must be 8. You FIRST think about the reasoning process as an internal monologue and then provide the final answer. The reasoning process and answer are enclosed within <think> </think> and <answer> </answer> tags. The answer consists only of several coordinate points, with the overall format being: <think> reasoning process here </think><answer><point>[[x1, y1], [x2, y2], ..., [x8, y8]]</point></answer>.

---

## D In-depth Analysis based on Real-world Experiments

**Table 9:** Real-world Execution Time (s)

| Task | Embodied-R1 | OpenVLA (FT) | MOKA | FSD |
|---|---|---|---|---|
| Move the sponge | **10s** | 23s | 28s | 14s |
| Pick up the cucumber | **8s** | 12s | 18s | 10s |

**Real-World Execution Latency.** We benchmarked the real-world execution latency of Embodied-R1 against various model archetypes, including the end-to-end OpenVLA (O'Neill et al., 2023), the modular MOKA (Liu et al., 2024a), and the affordance-based FSD (Yuan et al., 2025), with results reported in Tab. 9. Although Embodied-R1 employs a reason-then-execute process, it achieves the

fastest performance. This efficiency stems from its single-pass inference approach, where reasoning is performed only once before execution begins. In contrast, OpenVLA requires step-by-step inference and mandatory fine-tuning for new tasks, while modular methods like MOKA incur substantial system overhead due to their multi-component pipelines. Furthermore, among reasoning-based models, the 3B-parameter Embodied-R1 outperforms the 13B-parameter FSD, demonstrating superior inference efficiency due to its smaller size. Therefore, Embodied-R1 strikes an effective balance between powerful zero-shot manipulation capabilities and low real-world latency.

**Comparison with the A0 Model.** We compare Embodied-R1 to the A0 (Xu et al., 2025) model, which also generates object-centric visual traces but is trained exclusively through Supervised Fine-Tuning (SFT) and lacks a reasoning process. In our zero-shot generalization experiments, we found that despite extensive tuning efforts, A0 consistently failed to generate reliable trajectories, resulting in a success rate approaching zero. As shown in Fig. 7, we present a side-by-side visual comparison of the trajectories generated by Embodied-R1 and A0 in our real-world robot experiments.

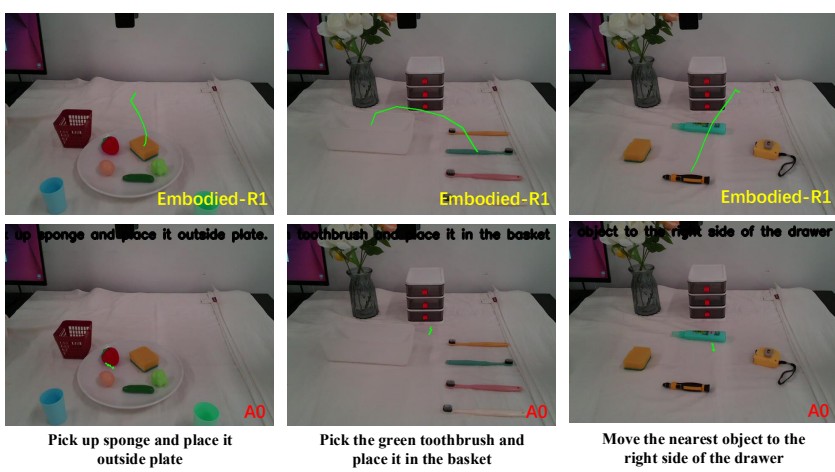

**Figure 7: Comparison of Visual Trace Generation between Embodied-R1 and A0 Models in Real-World Experiments**

**Failure Case Analysis.** Here, we provide an analysis of the failure modes for the results in Tab. 5. The MOKA pipeline exhibited low success rates across the majority of tasks. Its failures were often due to a cascade of errors originating from multiple stages: incorrect object recognition, failure to select the correct keypoints, or erroneous reasoning from the GPT module, which collectively led to extremely poor performance. Furthermore, even when MOKA did succeed in grasping an object, it struggled with accurate placement. For RoboPoint and FSD, the core issue was their extremely low success rate on tasks that require reasoning, such as "find the nearest object" or "pick the toothbrush of the correct color." Additionally, these models suffered from limited pointing accuracy, leading to low success rates when attempting to grasp challenging objects. In contrast, Embodied-R1 possesses robust spatial reasoning and precise pointing capabilities, allowing it to perform exceptionally well on most tasks and overcome the limitations of the other methods.

# E    MORE VISUALIZATIONS

Below, we provide visualizations of the real-robot execution process and the VTG tasks. For additional video materials, please see our project website: https://embodied-r1.github.io/

**More Visualization of Real World Manipulation Process** We showcase the process of Embodied-R1 performing real-world tasks in Fig. 8. To further evaluate its robustness, we visualize the process of Embodied-R1 performing Task 6 under different visual disturbances in Fig. 9.

**More Visualization of VTG Task** We provide additional visualization examples of Embodied-R1's predicted visual traces in Fig. 10. It can be seen that Embodied-R1 achieves accurate visual trajectory prediction across various scenarios.

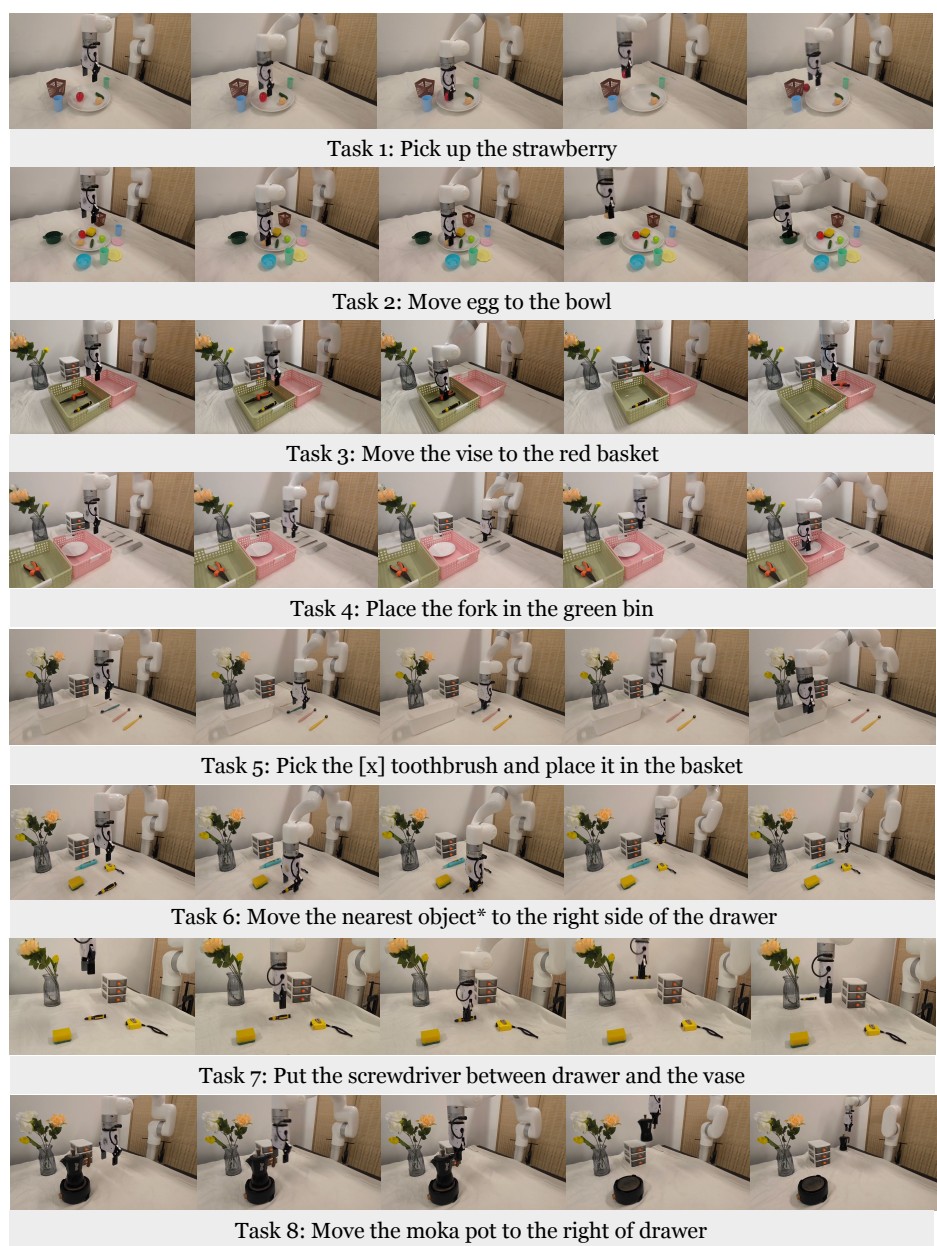

Task 1: Pick up the strawberry

Task 2: Move egg to the bowl

Task 3: Move the vise to the red basket

Task 4: Place the fork in the green bin

Task 5: Pick the [x] toothbrush and place it in the basket

Task 6: Move the nearest object* to the right side of the drawer

Task 7: Put the screwdriver between drawer and the vase

Task 8: Move the moka pot to the right of drawer

**Figure 8:** **The process of Embodied-R1 performing real-world tasks.**

## F    ADDITIONAL EXPERIMENTS

**The Phenomenon of Reward Hacking in VTG Tasks** We carefully designed the reward function so that the reward for each task is only related to the final goal, thereby avoiding reward hacking caused by intermediate rewards. However, we found that in the VTG task, designing the reward solely based on the distance between the predicted trajectory and the target trajectory can still result in reward hacking. The model quickly learned that in visual trajectory generation tasks, accurately predicting the starting and ending points is both crucial and relatively easy, leading it to converge rapidly to outputs with only these two points while ignoring the generation of intermediate trajectory points. We observed that by forcing the model to output multiple points and applying reward constraints for format reward, it becomes possible to generate complete visual traces. Therefore, we explicitly require the model to output a visual trace with eight points; otherwise, all rewards are set to zero

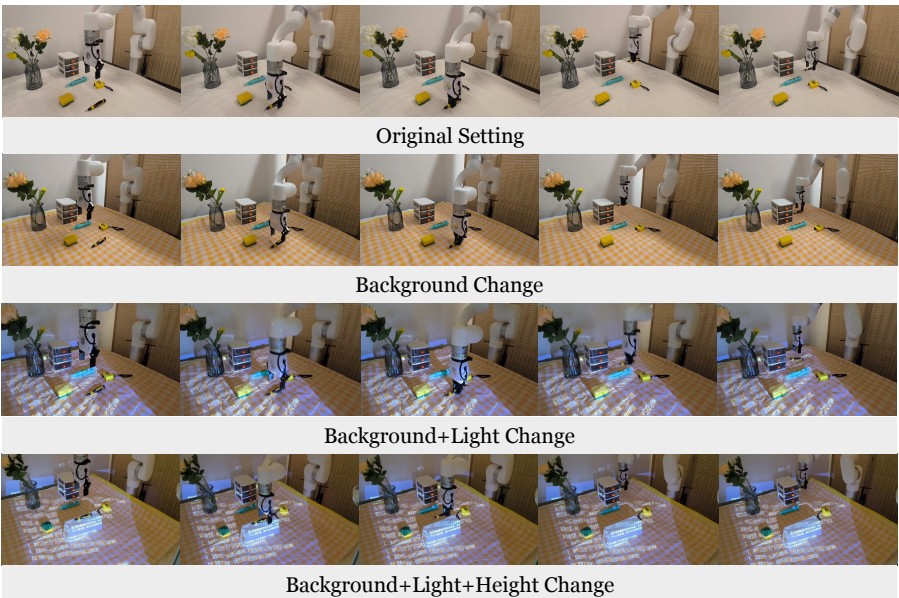

**Figure 9: The process of Embodied-R1 performing Task 6 under different visual disturbances.**

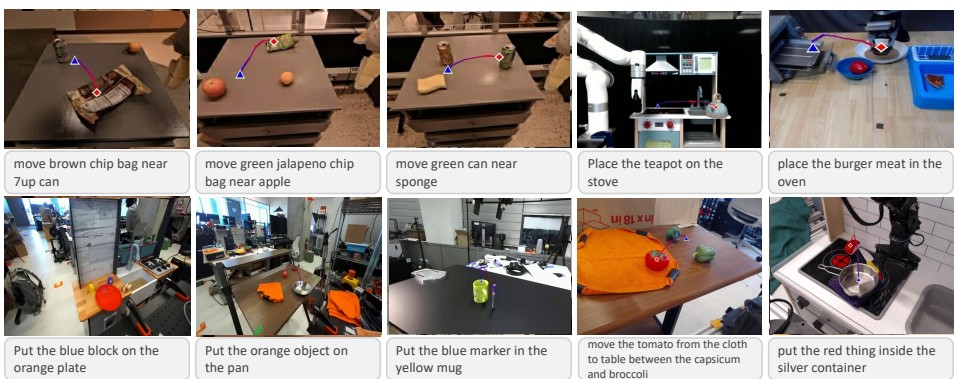

**Figure 10: Visualizing Embodied-R1's Prediction on VTG Tasks across Various Scenarios**

(since the format reward is not satisfied, subsequent analysis will not be performed). As shown in Tab. 10, we present the results on VABench-V, demonstrating the performance differences with and without the trajectory point number constraint. After modifying the reward function, the model is better able to fit the visual trace, achieving lower RMSE and a higher GPT score.

**Table 10:** Comparasion of w/ and w/o Point Num Reward. **Bolds** are better.

| VABench-VisualTrace | RMSE↓ | MAE↓ | GPT Score↑ |
|---|---|---|---|
| w/ Point Num Constraint | **77.83** | **44.97** | **7.27** |
| w/o Point Num Constraint | 105.2 | 59.7 | 5.57 |

**Qualitative Comparison: Embodied-R1 vs. SFT** To provide a deeper insight into the performance gains of our model, we conducted a qualitative analysis comparing Embodied-R1 with the SFT baseline. As illustrated in Fig. 11, the difference in capabilities is stark. In the first task, "move the orange toy inside the right sink in the bin," Embodied-R1 first articulates a clear plan: it identifies the toy's initial position, determines the need to move it towards the sink's center, and then guides it into the bin. This logical reasoning translates into a precise and successful visual trajectory. The SFT baseline, however, produces an erroneous trajectory that fails to place the toy correctly. Similarly,

for the second task, "place blue chip bag into white bowl," Embodied-R1 correctly reasons that the task requires moving to the bag, lifting it, positioning it over the bowl, and then lowering it. This step-by-step plan underpins the generated trajectory, which successfully completes the task. The SFT model again fails, generating a flawed path that does not accomplish the goal. These examples strongly suggest that the explicit chain-of-thought reasoning in Embodied-R1 is critical for understanding complex instructions and planning accurate, multi-stage visual trajectories, a capability that the standard SFT approach evidently lacks.

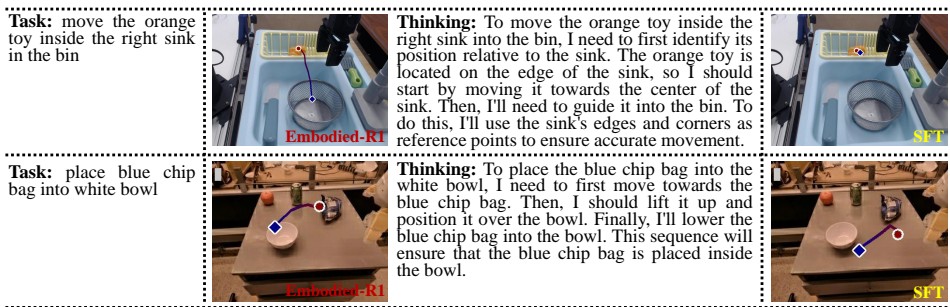

**Figure 11: Qualitative comparison of Embodied-R1 and the SFT baseline**. Our model Embodied-R1, leverages chain-of-thought reasoning (middle column) to generate a successful visual trace (left column). In contrast, the SFT baseline, which lacks an explicit reasoning process, produces incorrect trajectories (right column) for the same tasks.

**Table 11: Comparison of Mixed Training of Multiple Datasets and Only Training Corresponding Dataset.**

|  | Mixed Training | Unmixed Training |
|---|---|---|
| Part-Afford | **56.63** | 51.25 |
| Where2Place | **69.50** | 65.50 |
| VABench-P | **66.00** | 65.39 |

**Advantage of Mixed Training** We performed multi-task joint training of the various abilities required for embodied pointing and reasoning in the second stage. The advantage of this approach is that all abilities can share general knowledge of point coordinates and semantic space alignment during training, thereby achieving better performance. To validate this idea, we conducted multiple sets of experiments comparing the performance of mixed training and unmixed training. In the unmixed training setting, only the data corresponding to the benchmark ability is used; for example, training for one epoch only on the HandAL dataset and then testing on the Part-Afford dataset. As shown in Tab. 11, joint training consistently improves the success rate across multiple tasks compared to unmixed training. We believe that mixed training enables knowledge sharing among multiple abilities, enhances semantic understanding of spatial information, and thus leads to better generalization.

**Case Analysis of Embodied-R1** As shown in Fig. 12, we demonstrated the reasoning pathways of Embodied-R1 when facing different tasks. Even without any SFT stage, Embodied-R1 exhibited a human-like and rational reasoning process: it first infers the target object to focus on based on the task goal, then analyzes the relative spatial relationship between the object and the surrounding environment, and subsequently performs step-by-step reasoning to determine the target location (in RRG and VTG tasks), strictly adhering to a structured process of reasoning before providing the final answer. In some cases, Embodied-R1 still demonstrates a clear and accurate reasoning process even when confronted with cluttered scenarios.

**Integration with Learning-Based Pipeline.** To demonstrate the potential of Embodied-R1 in dynamic settings, we significantly expanded our evaluation to the full LIBERO benchmark suites (Spatial, Object, Goal, and Long, a total of 40 tasks). Each task has 50 demonstrations. We benchmarked the integration of Embodied-R1 with Diffusion Policy (Chi et al., 2023) based on the CleanDiffuser (Dong et al., 2024a) codebase, where Embodied-R1 serves as an efficient conditional planner. Specifically, it synthesizes a complete visual trace based on the initial observation and language instruction, which serves as a dense spatiotemporal condition to guide the downstream

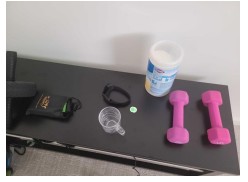
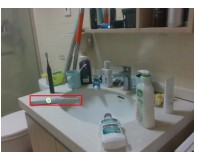
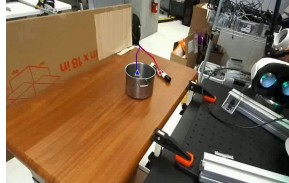

Q: Identify some points within the free space that is beside the glass cup and to the left of the weights.

*<think>*To identify points in the free space beside the glass cup and to the left of the weights, I need to consider the layout of the objects on the table. **The glass cup is placed on the table, and there are two dumbbells to the right of it.** The free space would be the area between these objects. However, since the question asks for points in the free space, I should focus on **identifying areas where no objects are present.** This could include the entire surface of the table or any gaps between the objects.*</think>*
*<answer><point>[[350, 298], [346, 297]]</point></answer>*

Q: Provide one or more points coordinate of objects region this sentence describes: I would like to pick up the white electric toothbrush.

*<think>*The white electric toothbrush is **located on the left side of the sink, closer to the edge of the counter**. It appears to be **lying horizontally on the countertop**.*</think>*
*<answer><point>[[156, 304], [158, 307]]</point></answer>*

Q: Put the marker in the pot. Use 2D points to mark the manipulated object-centric waypoints to guide the robot to successfully complete the task.

*<think>*To put the marker in the pot, I need to first locate the marker and the pot. The marker is likely on the table or nearby, while the pot is on the table as well. **I will move towards the marker and then position it inside the pot.** This requires me to **move from the marker's location to the pot's location**.*</think>*
*<answer><point>[[660, 300], [654, 287], [639, 260], [612, 240], [598, 230], [586, 250], [586, 280], [589, 298]]</point><answer>*

Figure 12: **Case Analysis:** Embodied-R1 possesses embodied reasoning capabilities. It can progressively locate relevant objects and infer spatial relationships according to task instructions, and ultimately provide coordinates through pointing based on embodied scene analysis.

policy. Since LIBERO-Long involves the composition of tasks in two stages, the agent queries the Embodied-R1 model every 15 steps to update the visual trace. As detailed in Tab. 12, this integration yields a substantial performance uplift, boosting the average success rate from 64.4% (DP Baseline) to 79.4% (DP + Embodied-R1), achieving an improvement of 15%.

We attribute this gain to the nature of the guidance: compared to the sparse, goal-oriented signals from language instructions, the visual trace provides explicit, dense path constraints. This helps the policy disambiguate instructions and maintain long-horizon consistency, validating that Embodied-R1 can effectively bridge high-level reasoning with low-level learning-based control.

Table 12: **Performance on the full LIBERO benchmark suite**. Integration with Embodied-R1 yields significant improvements across all task categories compared to the DP baseline.

|                   | LIBERO Goal | LIBERO Spatial | LIBERO Object | LIBERO Long | Avg |
|-------------------|-------------|----------------|---------------|-------------|-----|
| DP Baseline       | 64.3        | 76.1           | 71.4          | 45.9        | 64.4 |
| DP + Embodied-R1  | **80.7**    | **89.6**       | **78.7**      | **68.7**    | **79.4 (↑15%)** |

**3D RRG Capability of Embodied-R1** To explore our framework's capabilities to 3D spatial reasoning, we developed an RGB-D variant, named Embodied-R1-RGBD, utilizing synthetic paired data from YCB (Xiang et al., 2017) and ObjaverseXL (Deitke et al., 2023). Addressing the challenge of modal interference often found in direct concatenation, we implemented a Dual-Tower architecture (Zhou et al., 2025a). In this design, independent visual encoders process the RGB and depth inputs separately, ensuring that geometric depth cues are distinctively preserved before fusion. We further devised a two-stage training strategy to ensure precise 3D understanding: (1) Modality Alignment: We first conducted SFT on the large-scale RefSpatial (Zhou et al., 2025a) dataset (1M samples) to align the depth encoder's feature representation with the embedding space of LLM; (2) Reasoning Fine-Tuning (RFT): We then performed full-parameter fine-tuning to optimize the model specifically for 3D RRG tasks. Evaluation on the Open6DOR-Position benchmark (Qi et al., 2025) demonstrates the effectiveness of this approach. As shown in Tab. 13, Embodied-R1-RGBD achieves an overall accuracy of 94.3%. Notably, it excels in handling complex spatial relations (Level 1), scoring 74.5%, and achieves 99.2% on basic tasks (Level 0). These results confirm that our designs enable the feasibility of 3D RRG tasks, thereby validating the effectiveness of our paradigm's scalability to 3D. We believe that further design and adaptation are still required to extend the pointing paradigm to more 3D tasks and broader scenarios, but its future feasibility is highly promising. This will constitute a key direction for our subsequent research.

**Table 13: Performance on Open6DOR-Position Benchmark**

| Benchmark | *Level0* | *Level1* | Overall |
|---|---|---|---|
| GPT-4V | 46.8 | 39.1 | 45.2 |
| Qwen2.5-VL Bai et al. (2025b) | 59.5 | 36.2 | 54.9 |
| VoxPoser Huang et al. (2023) | 35.6 | 21.7 | 32.6 |
| SoFar Qi et al. (2025) | 96.0 | **81.5** | 93.0 |
| Embodied-SFT | 62.4 | 44.7 | 58.9 |
| Embodied-R1-RGB | 68.5 | 59.4 | 66.8 |
| Embodied-R1-RGBD | **99.2** | 74.5 | **94.3** |

## G   LIMITATION AND FUTURE WORK

Despite the state-of-the-art performance achieved by Embodied-R1 across numerous benchmarks and real-world tasks, this work has several limitations that present avenues for future research.

- **Potential for Integration with Learning-based Policies.** Our current approach primarily pairs the perception and reasoning capabilities of Embodied-R1 with a classical motion planner. A promising future direction is to integrate the model as a high-level front-end for a learning-based conditional policy, which promises to enhance execution efficiency and reactivity in dynamic environments. While several studies Bharadhwaj et al. (2024); Gu et al. (2023); Xu et al. (2024) have explored conditioning policies on visual traces to improve performance, these works focus on downstream policy design and do not provide a general-purpose visual trace predictor. We present preliminary experiments in App. F where a simple integration with Diffusion Policy conditions the strategy on visual traces. Notably, Embodied-R1 improves performance on LIBERO tasks without any additional fine-tuning. Therefore, this integration path holds significant potential and warrants further exploration.

- **Untapped Potential in Long-horizon Tasks.** The current framework is designed to process single-step instructions and does not natively include a mechanism for decomposing long-horizon commands (e.g., "prepare a meal"). However, this could be addressed through a modular, hierarchical approach. Embodied-R1 is well-suited to act as a robust execution module for individual sub-tasks. A high-level embodied planner could first decompose a complex instruction into a sequence of simpler steps, which would then be passed to Embodied-R1 for execution, enabling the system to tackle complex, multi-stage problems.

- **Inherent Limitations of the "Pointing" Representation.** While the pointing representation is effective for localization, placement, and trajectory generation, it may be insufficient for the full spectrum of complex robotic manipulation. Tasks requiring precise force control, twisting, wiping, or intricate interactions with deformable objects demand a richer representation than 2D coordinate points. We believe this issue can be mitigated by coupling the high-level "pointing" commands with a learnable downstream policy that can translate these targets into complex, dynamic actions. The design for Embodied-R1 reflects our primary focus on providing a promising solution for zero-shot generalization, for which a simplified, embodiment-agnostic intermediate representation is a key advantage.

- **Preliminary Integration of 3D Information.** The exploration of an RGB-D version of the model is still in its early stages. The paper notes that in tasks with complex spatial relations, the performance of the RGB-D variant can be slightly lower than its 2D counterpart. It is hypothesized that "depth map understanding may be more prone to hallucinations," indicating that robustly fusing 3D information into the model requires further development (Zhou et al., 2025b) .

- **Physical Common-Sense Alignment.** Another highly promising direction for future work is to mitigate physical common-sense hallucinations in embodied reasoning, such as proposing physically impossible interactions or ignoring fundamental physical laws like gravity. We plan to adopt Reinforcement Learning from Human Feedback (RLHF) (Dong et al., 2024b; Yuan et al., 2024c; Liu et al., 2024b) applied to the intermediate thinking process of the model, forcing the model's cognitive process to align with human intentions.

# H In-depth Comparison with Previous Works

To explicitly delineate the novelty of our "Pointing" paradigm, we provide a comprehensive comparison with representative state-of-the-art baselines: **RoboPoint** (Yuan et al., 2024b), **RoboBrain** (Ji et al., 2025), **A0** (Xu et al., 2025), and **FSD** (Yuan et al., 2025). Detailed comparisons across representation types, capability scopes, and training methodologies are presented in Tab. 14. The fundamental distinctions are analyzed across three key dimensions below.

**Table 14: Detailed Comparison of Embodied-R1 with Prior Point-based VLA Methods.** Robot-Trace refers to robot-centric visual trace, while Obj-Trace refers to object-centric visual trace.

| Feature | RoboPoint | RoboBrain | A0 | FSD | Embodied-R1 (Ours) |
|---|---|---|---|---|---|
| *1. Visual Representation & Embodiment* | | | | | |
| Visual Aids Type | Point | BBox / Robot-Trace | Obj-Trace | BBox / Point / Obj-Trace | **Point / Obj-Trace** |
| Embodiment-Agnostic | Yes | No (Robot-centric) | Yes | Yes | **Yes** |
| *2. Capability Scope* | | | | | |
| REG | Yes (Point) | Yes (BBox) | No | Yes (BBox/Point) | **Yes (Point)** |
| RRG | Yes | No | No | Yes | **Yes** |
| OFG | No | No | No | No | **Yes** |
| VTG | No | Yes (Robot-Trace) | Yes (Obj-Trace) | Yes (Obj-Trace) | **Yes (Obj-Trace)** |
| *3. Methodology & Reasoning* | | | | | |
| Training Method | SFT | SFT | SFT | SFT | **RFT** |
| Thinking Process | No | No | No | Yes (Fixed CoT) | **Yes (Free-form)** |
| Spatial/Logical Reasoning | N/A | N/A | N/A | Weak | **Strong** |

**Unified Interface vs. Fragmented Capabilities.** While geometric points are a common primitive, prior methods utilize them in fragmented or restrictive ways. Specifically, A0 focuses exclusively on object-centric visual trace (VTG) but lacks the semantic understanding required for grounding (REG) or functional analysis (OFG). RoboPoint limits points strictly to spatial constraints (REG, RRG). RoboBrain relies heavily on bounding boxes, which are often too coarse for precise manipulation (potentially causing operation offsets), and utilizes robot-centric traces that limit cross-embodiment transferability. In contrast, Embodied-R1 eschews coarse bounding box signals. Instead, it innovatively unifies four core capabilities (REG, RRG, OFG, VTG) into a single, embodiment-agnostic point-based interface (where visual traces are formulated as ordered sets of points). This unification allows Embodied-R1 to function as a "Generalist Pointing Agent," capable of handling diverse manipulation requirements within a single model architecture.

**Embodied-R1 acquires superior comprehension and reasoning abilities through RFT reasoning.** A critical distinction lies in the acquisition of comprehension and reasoning abilities. Baselines such as FSD, RoboPoint, and A0 rely primarily on SFT. Models lacking an explicit reasoning process (e.g., RoboPoint, A0) are limited to simple, explicit instructions (e.g., *"point to the right of the plate"*) and fail at tasks requiring multi-step deduction (e.g., *"pick up the object nearest to the robot"*). While FSD incorporates reasoning, it is constrained by fixed CoT templates derived from SFT data. As evidenced in our real-world experiments in Tab. 5, FSD suffers from low success rates in tasks requiring flexible spatial understanding or logical reasoning due to this rigidity in OOD scenarios. Conversely, Embodied-R1 employs RFT to develop free-form internal thinking and active spatial/logical reasoning. As shown in Fig. 6, this paradigm enables the model to robustly handle long-horizon and reasoning tasks.

**Multi-Task Synergy and Efficiency.** By unifying these different capabilities into a single "Pointing" task during the RFT stage, Embodied-R1 achieves outstanding coordinate understanding and semantic alignment with only 3 billion parameters. Significant performance improvements are observed across more than ten benchmarks, including tasks of spatial understanding and pointing. The ablation studies presented in Tab. 11 also demonstrate that unified multi-task modeling and learning bring positive improvements.

# I Use of LLM

We utilized LLMs as a writing assistance tool during the preparation of this manuscript. The use of LLMs was strictly limited to polishing the text, which included improving grammar, refining sentence structure, and enhancing overall clarity and readability. The core research concepts, methodologies, and conclusions were developed entirely by the authors.

