# OpenReview forum: "Embodied-R1: Reinforced Embodied Reasoning for General Robotic Manipulation"
_ICLR.cc/2026/Conference — ICLR 2026 Poster_

### Official Review · Reviewer_LhSQ · 2025-10-27

**Soundness:** 2
**Presentation:** 3
**Contribution:** 1
**Rating:** 2
**Confidence:** 5

**Summary:**

In this paper, the author proposed a pipeline of RL post-training of VLM models to perform embodied related perception tasks. The core innovation is "pointing" as a unified, embodiment-agnostic intermediate representation that bridges high-level vision-language comprehension with low-level action primitives. The authors define four embodied pointing abilities: Referring Expression Grounding (REG), Region Referring Grounding (RRG), Object Functional Grounding (OFG) and Visual Trace Generation (VTG). To train the model, the authors curate Embodied-Points-200K, a 200,000-sample dataset structured as "question-verification" pairs to handle the multi-solution nature of pointing tasks. It draws from diverse sources like RefCOCO, RoboRefIt, HandAL, and synthetic Isaac Gym simulations, augmented by GPT-4o for question rewriting.

**Strengths:**

1. The paper is well-written and easy to follow.
2. The evaluation benchmarks are comprehensive.
3. The proposed RL method improves the performance of zero-shot reasoning tasks.

**Weaknesses:**

1. Lack of novelty. For the whole pipeline, I think it is highly similar to the Seg-Zero [1] and VisionReasoner [2]. The authors finetuned Qwen2.5-VL-3B using Reinforced Fine-tuning (RFT) approach, closely mirroring the training paradigms introduced the two papers. From my perspective, the authors essentially repurposed the reasoning detection, segmentation, and point prediction tasks from prior papers, adapting them into object-manipulation-level pointing prediction tasks. Apart from introducing a few additional reward functions, the overall framework remains fundamentally unchanged. Most importantly, I noticed that the authors did not cite these two papers at all, despite the fact that their publicly released anonymous code is highly similar to the open-source implementations of those two papers. I would like to remind the authors of the principle of academic integrity. Also, treating point or affordance as a unified representation is discussed broadly in previous works.

2. Insufficient literature review. Several highly similar works are neither compared nor discussed, such as Affordance-R1 [3].

3. Lack of comparisons of scalability. What would happen if you scale up the model size from 3B to 7B or larger ones?

4. Meaningless comparisons. In the VLAs experiments, the authors compared the Embodied-R1 directly with VLAs, where the Embodied-R1 outputs the point prediction and utilizes another planner to solve the path planning. However, the VLAs directly output all-step actions in relatively high control frequency, which is intrinsically more difficult to complete manipulation tasks.  Hence, such kind of comparisons are meaningless from my perspective.

5. Insufficient experiments. How about the dual-arm experiments, does the Embodied-R1 perform good?

6. Overall, I think the paper's motivation is not strong enough, that we know the RL post-triaing strengthens the perception capabilities as the authors claimed. However, how to prove such boost of performance will lead to improvements of real manipulation tasks? In other words, the object-level pointing paradigm is limited compared to the end-to-end VLA paradigm, which I think limits the meaning of this paper. To be more specific, the claimed paradigm and model are useless in manipulation tasks that are not object-centric.

[1] Seg-Zero: Reasoning-Chain Guided Segmentation via Cognitive Reinforcement

[2] VisionReasoner: Unified Reasoning-Integrated Visual Perception via Reinforcement Learning

[3] Affordance-R1: Reinforcement Learning for Generalizable Affordance Reasoning in Multimodal Large Language Model

**Questions:**

See the weakness.

---

> ### Public Comment · ~Yuqi_Liu4 · 2025-11-18
> **Public Comment for Embodied-R1**
>
> Thank you for your time and efforts. The works you mentioned appear to be concurrent studies and primarily focus on different application domains. Therefore, we believe a direct comparison may not be appropriate.

---

> > ### Comment · Reviewer_LhSQ · 2025-11-21
> > **Discussion with the public comment**
> >
> > Thank you for your public comment.
> >
> > With all due respect, I disagree with your points.
> >
> > First, I didn't ask for a direct comparison. What I said was that when a work heavily draws from or borrows ideas from another, it is problematic to make absolutely no mention or citation of the original work.
> >
> > Second, I don't think these are completely different fields. The tasks performed are exactly the same—both involve bbox inference and similar tasks—the only difference is the source of the data.
> >
> > Finally, I don't consider it concurrent work. According to ICLR's review policy, only papers with submission deadlines after July 24, 2025 can be treated as concurrent work.

---

> ### Author Response · Authors · 2025-11-25
> **Rebuttal for Reviewer LhSQ [1/5]**
>
> We sincerely thank you for your detailed review and valuable feedback. We take your comments very seriously. However, regarding the concerns raised about **"lack of novelty," "high code similarity (academic integrity),"** and **"significance of comparison,"** we believe there are significant misunderstandings.
> **We believe it is crucial to respectfully but firmly object to the concerns regarding academic integrity and clarify these points.** To resolve these doubts, we provide a point-by-point response below to demonstrate the independence, innovation, and unique contributions of our work to the Embodied AI community.
>
> ## **Q1. Clarification on Novelty (Comparison with Seg-Zero and VisionReasoner)**
>
> ### A. Clarifying the "Borrowed Ideas" Misconception
>
> Our research is entirely rooted in the Embodied AI research path, specifically **the Affordance VLA direction in Embodied AI**, rather than general vision perception tasks.
> **1. Research Context & Motivation:** As stated in our Introduction, the field of robotic manipulation is primarily divided into three paths: End-to-End VLA, Modular VLA, and Affordance VLA. Our research is situated within the Affordance VLA paradigm. In this path, utilizing visual auxiliary signals (e.g., detection, segmentation, point prediction) to enhance robotic manipulation is the core paradigm. We detailed the development of this field in Section 2, "Visual Auxiliary Signals for Robotic Manipulation." While using points or affordances as a representation has been discussed in academia, our core contribution lies in the deep unification and optimization of this concept specifically for robotic manipulation.
> **2. Core Breakthroughs of Embodied-R1 Compared to Prior Works (e.g., RoboPoint[1], FSD[2], RoboBrain[3]):**
>
> - **Unified "Universal Pointing" Interface:** Previous methods utilized visual signals in a fragmented manner with limited application scenarios. For instance, RoboPoint only uses points for spatial constraints (REG/RRG), and RoboBrain relies mainly on bounding boxes for localization, which are often too coarse for precise manipulation. Furthermore, their robotic-centric visual trace limit generalization across different robot morphologies. Embodied-R1 discards coarse bounding box signals and innovatively unifies four core embodied capabilities—**Referring Expression Generation (REG), Referring Relation Generation (RRG), Object Function Grounding (OFG), and Visual Trajectory Generation (VTG)**—into a single **Embodiment-Agnostic Pointing Interface**. This unified design makes Embodied-R1 a universal pointer, satisfying diverse manipulation needs within a single model.
> - **Introducing RFT to Solve the Multi-Solution Dilemma & Reasoning Bottleneck:** RFT is used to mitigate the multi-solution problem in pointing tasks and endow the model with reasoning capabilities.
>     - **Current Limitations:** Models lacking reasoning processes (e.g., RoboPoint, A0[4]) can only handle simple, explicit instructions (e.g., "point to the right of the plate") and fail at complex logic. While FSD uses CoT, it is constrained by fixed template structures, performing poorly on OOD tasks requiring spatial understanding or multi-step logic.
>     - **Our Innovation:** Through RFT, Embodied-R1 develops **Free-form Thinking** and active spatial/logical reasoning capabilities. This allows it to robustly handle complex instructions (e.g., "pick up the object closest to the robot") and reasoning-intensive tasks (please refer to **Table 5 and Figure 6** in the paper for specific performance gains).
> - **Efficiency & Performance:** Thanks to carefully designed data construction, multi-task reward functions, and co-training strategies, with only **3B parameters,** Embodied-R1 significantly outperforms prior strong baselines across extensive evaluations, including benchmarks for 10+ spatial/pointing capabilities, 8 real-world tasks, and 4 simulation tasks.

---

> > ### Comment · Reviewer_LhSQ · 2025-11-27
> > **Response to authors**
> >
> > With all due respect, I disagree with your point.
> >
> > The definition of VLA is *Definition I.1 (Vision-Language-Action (VLA) Model). A
> > Vision-Language-Action (VLA) model is a system that takes
> > visual observations and natural language instructions as
> > required inputs and may incorporate additional sensory
> > modalities. It produces robot actions by directly generating
> > control commands. Thus, models in which a high-level policy (e.g., a vision-language model backbone) merely selects
> > an index from a set of pre-trained skills or control primitives
> > are excluded from this definition.* according to [1]. Based on both the academic definition and my personal understanding, the Embodied R1 has no relation with the VLA, as the Embodied R1 does not output the actions **directly** and in an **end-to-end** way. Instead, Embodied R1 outputs points for motion planner or using IK to solve the joint pos. In another word, It  cannot output dense actions in e2e and high-frequency way. Therefore, I disagree with you on that you concluded the Embodied R1 into the "Affordance VLA" class.
> >
> > [1] Kawaharazuka, K., Oh, J., Yamada, J., Posner, I., & Zhu, Y. (2025). Vision-language-action models for robotics: A review towards real-world applications. IEEE Access.

---

> > > ### Author Response · Authors · 2025-11-27
> > >
> > > Dear Reviewer LhSQ,
> > >
> > > Thank you for your valuable feedback and for highlighting the precise definition of Vision-Language-Action (VLA) models as referenced in Kawaharazuka et al. (2025).
> > >
> > > ### **Classification Logic and Clarification:**
> > >
> > > We realize the source of the confusion regarding our classification. Our categorization of "End-to-end VLA," "Modular VLA," and "Affordance VLA" was primarily adapted from the classification used in the **A0[1]** paper. **In our original submission, we adopted a broad definition where "VLA" referred to any comprehensive system that takes vision and language as input and ultimately results in robotic action.** Our use of the term "Affordance VLA" was intended to group methods that utilize VLAs/VLMs as the core reasoning backbone for robotic manipulation.
> > >
> > > ### **Acknowledgment of Strict Definition:**
> > >
> > > We recognize that this broad usage may cause confusion across different research backgrounds. We fully acknowledge that under the strict academic definition provided in Kawaharazuka et al. (2025), a VLA model is characterized by the direct generation of control commands from visual and language inputs. Since Embodied-R1 generates intermediate representations (points/trajectories) that are subsequently processed by a motion planner, it is indeed more accurately classified as a Affordance methods/agents rather than an End-to-End VLA.
> > >
> > > We have two ways to make modifications: one is to specify in our classification that VLA refers to models capable of directly generating control commands from visual and language inputs; the other is to replace "Modular/Affordance VLA" with "Modular/Affordance Methods.”
> > >
> > > ### **Action Plan for Revision:**
> > >
> > > We believe this can be addressed with precise terminology updates rather than large-scale structural changes. We have formulated a revision plan **(to be completed within one day)**:
> > >
> > > - **Terminology Update:** In the revised manuscript, we will change "Affordance VLA" and "Modular VLA" to **"Affordance-based Methods"** and **"Modular Methods"** respectively. This change will avoid potential misunderstanding and accurately reflect the architectural characteristics. All related descriptions will be updated accordingly.
> > >
> > > ### **Validity of Comparisons:**
> > >
> > > 1. **Primary Comparison:** The primary baselines in our paper remain similar affordance-based and modular methods. Our results demonstrate that Embodied-R1 surpasses these methods in both performance and broad applicability.
> > > 2. **Scientific Value of Cross-Paradigm Comparison:** While we accept the taxonomic distinction, we believe comparing Embodied-R1 against End-to-End VLAs (e.g., OpenVLA) on standard benchmarks remains scientifically critical. Both paradigms aim to solve the same "Image-to-Action" problem, and "Success Rate" serves as the gold standard for evaluation. This type of cross-paradigm comparison is standard practice in recent works like **SoFar[2]** (Modular) and **FSD** [3](Affordance). Our results highlight a key finding: a **Modular approach (Embodied-R1)** can achieve superior zero-shot OOD generalization compared to fine-tuned **End-to-End VLAs**, which is a significant advantage of our work.
> > >
> > > Thank you for your correction regarding taxonomy, which has improved the rigor of our paper. **Apart from the taxonomy issue, we would like to know whether our responses to the other key concerns have alleviated your worries. We are eager to discuss this with you further.**
> > >
> > > Best regards,
> > > The Authors
> > >
> > > ---
> > >
> > > [1] Xu et al. A0: An Affordance-Aware Hierarchical Model for General Robotic Manipulation. ICCV2025.
> > >
> > > [2] Qi et al. SoFar: Language-Grounded Orientation Bridges Spatial Reasoning and Object Manipulation
> > >
> > > [3] Yuan et al. From Seeing to Doing: Bridging Reasoning and Decision for Robotic Manipulation. Arxiv2025.

---

> ### Author Response · Authors · 2025-11-25
> **Rebuttal for Reviewer LhSQ [2/5]**
>
> ### B. Direct Comparison with Seg-Zero and VisionReasoner
>
> We now directly address the comparison with Seg-Zero and VisionReasoner.
> First, we acknowledge the technical commonalities: all three involve point coordinate prediction and utilize the RFT paradigm based on Qwen2.5-VL. It is important to note that in VLM-RL post-training, using mainstream open-source base models and fine-tuning frameworks is a standard community technical choice.
> However, we must emphasize that Embodied-R1 is fundamentally different from these works in **task domain, capability scope, and technical implementation**.
>
> - **Fundamental Difference in Motivation and Task Definition:** Our research is based on the **Affordance VLA** route, aiming to bridge the specific "Seeing-to-Doing" gap in robotic manipulation. In contrast, Seg-Zero and VisionReasoner belong to the **RL for Improving Perception** route.
> - **Different Objectives:** The competing methods predict points to serve as prompts for segmentation models (like SAM) for visual tasks (detection/segmentation). Embodied-R1 predicts points for physical interaction. **We pioneered the integration of REG, RRG, OFG, and VTG—four embodied pointing capabilities designed specifically for robotic grasping, placement, and trajectory planning, not simple image segmentation queries.**
>
> To substantiate this, we conducted comparative experiments on the REG and RRG benchmarks using the official VisionReasoner model (note: VisionReasoner does not support VTG tasks at all). The results are as follows:
>
> |  | RoboRefit (REG) | Where2Place (RRG) | VABench-P |
> | --- | --- | --- | --- |
> | VisionReasoner | 69.7 | 0.64 | 16.49 |
> | **Embodied-R1** | **85.58** | **69.50** | **66.00** |
>
> *Experimental results demonstrate that VisionReasoner's scope of capability is insufficient to support complex embodied intelligence tasks directly. This further proves that specialized "Embodied Pointing" research is absolutely necessary for embodied tasks.*
>
> - **Differences in Data and Rewards:** Based on the task differences above, the implementation details are completely different. Embodied-R1 constructed a specialized dataset containing RRG, OFG, and VTG data for the robotics domain, and designed a specialized reward function system (e.g., trajectory rewards, functional point distance rewards). These designs, tailored to embodied AI, are entirely absent in VisionReasoner, which is oriented toward general visual perception.
>
> ### C. Extended Related Work
>
> In the initial submission, we identified our primary related works as "Embodied Reasoning in Robotic Manipulation (including VLM + RL for Embodied AI)," "Spatial Reasoning with VLMs," and "Visual Auxiliary Signals for Robotic Manipulation (Including visual aids that assist in robotic decision-making)." We have established a new section: "Enhancing VLM Reasoning via Reinforced Fine-tuning." in the revision (**Appendix I, ADDITIONAL RELATED WORK).** We incorporate Seg-Zero, VisionReasoner and additional literature on improving visual perception/detection and segmentation in visual models using RFT, thereby enriching the comprehensiveness of our discussion.
>
> ## **Q2. Strict Clarification on Innovation and Academic Integrity**
>
> Building on the response to Q1, our research is fully grounded in Affordance VLA. We must now issue a **firm clarification regarding the accusation of "high code similarity." This is a factually incorrect allegation.**
>
> - **Vastly Different Research Domains:** As detailed in Q1, Embodied-R1 differs fundamentally from the cited works in task domain, capability scope, and technical implementation. Furthermore, all benchmark evaluations and robotic tasks in our paper are completely different from those in the cited works.
> - **The Truth About Code Similarity:** We meticulously checked the open-source code for Seg-Zero and VisionReasoner. The reality is that our project and these works share a model training framework based on **EasyR1[4]**, which is widely used in the open-source community. This is simply because EasyR1 is currently the most mainstream foundational framework for fine-tuning Qwen2.5-VL using RFT.
> - **Independent Implementation:** We have performed a **completely independent reconstruction** tailored to embodied tasks regarding data construction, dataloaders, reward function design, and the inference & evaluation pipeline. Using the same open-source training base does **not** constitute plagiarism.
>
> We have explicitly cited the source of the training framework in the "Implementation Details" section of the revised paper (see **Appendix B [The Embodied-R1 model is trained entirely based on the EasyR1 open-source codebase.]**) to ensure full transparency. Therefore, whether viewed from code implementation or research logic, Embodied-R1 is an independent work with high distinctiveness.

---

> > ### Comment · Reviewer_LhSQ · 2025-11-27
> > **Discussion about the Seg-zero and visionreasoner**
> >
> > I was **NOT** asking the authors to compare with these methods. Instead, I suggested that the authors should cite these highly-related paper. Borrowing ideas from other domains is quite common, but citation is also highly encouraged to show the respect to the authors of previous work.
> >
> > Regarding the authors’ claim that their approach is “fundamentally different from these works,” I would appreciate further clarification and discussion on the specific differences. I evaluate the novelty of a method primarily along three dimensions: model architecture, theoretical framework, and training objective. From this perspective, the proposed approach shares the same backbone (Qwen2.5-VL), the same theoretical foundation (GRPO-based reinforcement learning), and the same training objective (point-based prediction) as prior work. The only differences lie in the loss function and training data. Therefore, in my view, the method remains highly similar to existing approaches.
> >
> > **I would like to pose a direct question to the authors:**
> > If one takes VisionReasoner unchanged and simply trains it on the authors’ dataset using your proposed reward/loss function, would it achieve performance comparable to the method presented in this paper?
> > From my perspective, the answer is clearly yes. This is my major concern. If the authors can convincingly demonstrate why this would not be the case, I am willing to reconsider my score. Otherwise, this remains the primary reason for my current evaluation.

---

> > > ### Author Response · Authors · 2025-11-27
> > >
> > > Dear Reviewer LhSQ,
> > >
> > > We accept your suggestion and have expanded the expanded related work **"Improving VLM Reasoning via RFT,"** to incorporate and discuss VisionReasoner, Seg-Zero, and Affordance-R1, as you mentioned.
> > >
> > > We are now to provide a detailed clarification to demonstrate the novelty and contribution of our work in embodied AI applications. **Our paper is positioned in the "applications to robotics" track, addressing the core problem of enhancing zero-shot generalization** for general robotic manipulation across a broader range of tasks, alleviating this critical application challenge.
> > >
> > > We highly value your direct question: "If VisionReasoner is kept unchanged and only trained with the authors’ dataset and the proposed reward/loss functions, would its performance be comparable to the method proposed in this paper?" Our answer is **yes**, but with a crucial caveat: **this precisely constitutes the non-trivial and systematic contribution of our work in the domain of embodied robotics.**
> > >
> > > In the **"Applications to Robotics" track, the core challenge is not to develop new fundamental or general algorithms, but rather to successfully specialize and adapt a powerful general framework** to solve unique, critical application problems—in our case, achieving robust zero-shot generalization in robotic manipulation. Our entire methodology represents this successful specialization process, the value of which has been validated by achieving state-of-the-art performance in challenging real-world tasks.
> > >
> > > Our core contributions include the following:
> > >
> > > 1. **Innovative Capability Definition:** **Defining and implementing a unified, general embodied  pointing abilities**. Prior methods lacked comprehensive capabilities, restricting a single model to a limited range of tasks.
> > > 2. **Dataset Construction and Generation Supporting these Capabilities:** Achieving these comprehensive capabilities required meticulous design in determining what data to construct, how to generate the dataset, and what ratio to use for training. **We have also confirmed in the submitted version and this rebuttal that previous general VLMs (including GPT-4o and VisionReasoner, among others) lack the full scope of embodied capabilities,** leading to poor results.
> > > 3. **Reward Engineering for Robotics Challenges:** Designing an RFT reward function capable of stably supporting collaborative multi-task training, preventing reward hacking in VTG tasks, and addressing the multi-solution dilemma. Furthermore, we demonstrated that the mixed training of all four pointing abilities significantly outperforms non-mixed single-task training.
> > > 4. **Achieving Significant Generalization Performance Improvement in zero-shot robotic manipulation tasks, surpassing existing SOTA baselines**. Additionally, Embodied-R1 excels in tasks involving one-step or multi-step logical reasoning , while maintaining outstanding performance even when facing severe visual disturbances.
> > >
> > > ### **The Universal Paradigm of Applied Research**
> > >
> > > We want to emphasize that the difference lies not in individual components, but in the **systematic co-design** **of the capability definition, data construction, and multi-task RFT reward mechanism**, which is driven by the application needs. This is not uncommon for application-focused work.
> > >
> > > As you pointed out, the practice of using the same general foundation models (such as Qwen2.5-VL) and mainstream algorithms (GRPO)  in an application field, and solving specific problems through customized domain datasets and finely tuned reward functions, is a **common and effective paradigm** in current VLM application research.
> > >
> > > Indeed, we can observe that many excellent application works follow this pattern: for example, the VisionReasoner you mentioned builds upon Seg-Zero by expanding its capabilities and dataset; we can also find many other excellent works that use the Qwen2.5-VL backbone, are based on GRPO training, and modify the dataset and reward function to solve specific visual-language challenges, such as Vision-R1[1] and Video-R1[2].
> > >
> > > Therefore, adopting mainstream models and algorithms is standard practice in applied research. Embodied-R1 is a successful and non-trivial **systematic application and capability expansion** of the general RFT paradigm in the field of robotic manipulation. **All of our work and contributions are reflected in the comprehensive capability definition, dataset construction, multi-task collaborative reward design tailored for the robotics domain, and are ultimately validated by achieving SOTA performance in the most challenging zero-shot robotic tasks.**
> > >
> > > We hope this detailed explanation will convince you that our work represents a crucial step forward in the field of embodied robotics applications, possessing sufficient innovation and importance.
> > >
> > > [1] Reason-RFT: Reinforcement Fine-Tuning for Visual Reasoning of Vision Language Models. NeurIPS2025
> > > [2] Video-R1: Reinforcing Video Reasoning in MLLMs. NeurIPS2025

---

> > > > ### Comment · Reviewer_LhSQ · 2025-11-28
> > > > **Response to authors' commnet**
> > > >
> > > > To conclude our discussion, the authors admitted that 1. Embodied R1 differs from previous methods **only** in the aspects of training data and reward function. 2. The Embodied R1 is a perceptive model rather than VLA, fails to output the actions directly based on the vision-language input. 3. Negligently failing to cite some highly relevant papers (with identical model architecture and training methods). The authors' claim to novelty lies in successfully applying previous architectures and methods to robot-related perception tasks. I believe the transfer gap is not particularly large (since it involves making point-level predictions based on vision-language inputs), and therefore, in my opinion, the novelty is insufficient.
> > > >
> > > > I would suggest the authors to revise the submission based on these aspects. I will appropriately raise my score in the final stage, and I would very much like to have further discussion with the other reviewers and the AC to hear everyone's thoughts about my concerns.

---

> > > > > ### Author Response · Authors · 2025-11-28
> > > > >
> > > > > We sincerely appreciate your continued engagement and constructive suggestions. Based on our discussion, we have revised the manuscript to address potential misunderstandings and enhance its quality:
> > > > >
> > > > > **1. Terminology Correction:**
> > > > > We have replaced all expressions involving **“affordance/modular VLA”** with **“affordance/modular methods”** and highlighted these changes in red. **It is worth noting that this involved fewer than five edits,** as most descriptions in the original text already used terms like “methods.” In the introduction, we also explicitly stated that Embodied-R1 is an Embodied-VLM. These corrections were primarily in the “Type” column of the SIMPLEREnv experiment table and have now been fully resolved.
> > > > >
> > > > > **2. Expanded Review of VLM-RFT Work:**
> > > > > We conducted a more detailed survey on "Improving VLM Reasoning via RFT," moved this section **from the appendix to the main text, and added further references**.
> > > > >
> > > > > > “Building upon VLM foundation models, post-training with curated data using algorithms like GRPO (Shao et al., 2024) or PPO (Schulman et al., 2017) can achieve performance and generalization capabilities that surpass standard SFT. This methodology has been successfully applied across diverse domains, including affordance prediction (Wang et al., 2025a), video understanding (Park et al., 2025), and GUI agents (Zhou et al., 2025b). However, in the field of embodied AI, particularly in robotic manipulation, how to leverage RFT to fuse perception and decision-making to cover a wide range of task types remains to be further explored.”
> > > > > >
> > > > >
> > > > > In fact, **many application-oriented papers utilize Qwen2.5-VL and GRPO with identical architectures and training methods**, achieving excellent performance in their respective domains. We did not directly apply RFT for decision-making policy generation, but rather applied it to connect perception and decision-making, specifically filling this gap in the embodied domain.
> > > > >
> > > > > **3. Refined Contribution Statement:**
> > > > > We made minor adjustments to the contributions in the Introduction. We clearly state that our contribution stems from a comprehensive definition of capabilities. It lies not in a single component, but in the **systematic co-design of capability definition, dataset construction, and the multi-task RFT reward mechanism**, leading to significant improvements in application performance.
> > > > >
> > > > > **Finally, thank you for suggestions. But we would like to address some remaining misunderstandings.**
> > > > >
> > > > > - **Regarding "The only difference lies in training data and reward function":**
> > > > > **We respectfully suggest that this perspective overlooks the value of systematic contributions in most applied research.** Whether it is VLM+RFT work (e.g., VisionReasoner, Video-R1) or common Embodied VLM work (e.g., RoboPoint, RoboBrain), methodologically, **they can often be reduced to “changes in data or rewards.”** However, they provide immense value by driving application progress in their respective fields and becoming mainstream baselines. Our contribution is **systematic and holistic:** the unified and comprehensive definition of capabilities, the construction of required data (what is needed and how to build it), the design of multi-task RFT rewards, as well as large-scale experimental performance comparison and ablation analysis. Our aim is to provide a concise and efficient architecture and perspective for the training of Embodied VLM in the embodied AI community.
> > > > >
> > > > > - **Regarding "Embodied R1 is a perception model rather than VLA":**
> > > > > We acknowledge this point. Thank you for the correction; we have updated the classification in the SIMPLEREnv experiment table. However, our overall narrative logic and experimental comparisons remain valid, as our primary positioning and comparisons (in the vast majority of experiments) are against other Embodied VLM models / Affordance methods.
> > > > >
> > > > > - **Regarding "Essentially point-level predictions based on vision-language inputs":**
> > > > > We wish to emphasize that while previous baseline methods also performed point prediction, they were typically limited to single-model capabilities with a restricted task scope, and suffered from deficits in generalization and reasoning. Our core innovation lies in **defining and implementing a unified and general embodied pointing capability**, covering both static points and dynamic visual traces. Through this concise design and the synergy of these capabilities via RFT, our method can handle a much broader range of tasks. **This significantly improves success rates and generalization, enabling the handling of reasoning tasks and the application of visual robustness, thus breaking through the limitations of previous point prediction methods.** **We believe that being concise, effective, easy to apply, and free of redundant design is a key merit of Embodied-R1.**
> > > > >
> > > > > We hope these revisions and clarifications further improve the quality of the paper and fully address your concerns. We look forward to your further discussion.

---

> ### Author Response · Authors · 2025-11-25
> **Rebuttal for Reviewer LhSQ [3/5]**
>
> ## **Q3. Regarding Literature Review: Affordance-R1**
>
> Regarding Affordance-R1 [6], according to arXiv records, this paper was submitted in August 2025 (*Submitted on 8 Aug 2025 (v1)*). This is later than the ICLR concurrent work determination cutoff date (July 24, 2025). According to conference policy, this qualifies as **Concurrent Work**.
> Nevertheless, we have added it to the related work discussion in the revised version **(Appendix I, ADDITIONAL RELATED WORK).** It is worth noting that Affordance-R1, similar to other Affordance VLA works, focuses solely on the single capability of Object Function Grounding (OFG). It has limitations in handling tasks involving spatial relationship reasoning (RRG), such as "place the cup to the right of the plate," which highlights the advantage of Embodied-R1's multi-task unified paradigm.
>
> ## **Q4. Lack of Scalability Comparisons**
>
> We strictly focused on the 3B parameter scale to ensure deployability in the real world.
> - **Latency Constraints:** In real-world robotic applications, inference latency is critical. Scaling the model to 7B+ or 70B would significantly increase latency, making it difficult to meet the practical requirements of the high-flexibility interaction scenarios we target. Furthermore, the most mainstream and well-known embodied policy models (e.g., the **Pi series** and **Groot series**) are all around the 3B magnitude.
> - **Performance:** Despite being a 3B model, Embodied-R1 demonstrates excellent performance across multiple spatial understanding and embodied reasoning benchmarks, as well as in real-robot experiments, outperforming larger-scale models. This effectively promotes the democratization of embodied reasoning capabilities on edge devices.
>
> ## **Q5. Regarding Dual-Arm Experiment Results**
> - **Embodiment-Agnostic Nature:** Our method generates points/trajectories in the World or Camera coordinate system, which is inherently embodiment-agnostic. Downstream planners can easily map these semantic points to a dual-arm system (e.g., "Left arm goes to Point A, Right arm goes to Point B"). We focus on solving the Vision-Language bottleneck, leaving the kinematic solving to the planner—this is precisely the advantage of a modular design.
> - **Extensive Validation:** We have validated our method across **8 real-world tasks and 4 simulation suites across diverse embodiment and tasks**, including environments with long horizon and susceptibility to interference, demonstrating the model's robust generalization capabilities.
>
> ## **Q6. Response regarding the Comparison Embodied-R1 with End-to-End VLAs**
>
> We believe there may be some misunderstandings regarding the positioning and evaluation scope of our work.
>
> **1. Positioning and Primary Baselines (Affordance VLA)**
>
> First, **Embodied-R1 is explicitly positioned as an Affordance VLA**, with a primary focus on **zero-shot manipulation generalization**. Consequently, our comparative analysis centers on the most relevant **Affordance VLA and Modular VLA baselines** (specifically RoboPoint, FSD, MOKA[7], and SoFar[8]). **These baselines share a common paradigm with our work**: they output intermediate predictive results (affordances), which are then utilized by a separate planner for **open-loop decision-making**. Across all evaluations, ranging from spatial understanding and pointing benchmarks to real-world and simulation manipulation tasks, Embodied-R1 demonstrates performance far superior to these baselines.
>
> **2. Clarification on End-to-End VLA Comparisons**
> It is important to note that a direct comparison with End-to-End VLAs is not the central focus of this paper**.** Among our extensive results—spanning 10+ embodied QA benchmarks, 4 simulation suites, and 8 real-world tasks—**only the SIMPLEREnv results include comparisons with End-to-End VLAs.**
> Furthermore, these End-to-End VLAs were evaluated under their **standard experimental settings**. They operated in a standard **closed-loop** manner (receiving an observation, outputting action steps, and repeating the process until termination), rather than outputting all action steps at once as assumed.
>
> **3. Fairness of the Comparison**
> Since SIMPLEREnv is a standard benchmark specifically designed to test VLA generalization, all methods were tested under their optimal settings, and all methods received the same observational information. We believe this constitutes a fair comparison, as **Success Rate is the gold standard in robotic tasks.** Through extensive empirical evidence, analytical experiments, and comparisons against multiple strong baselines, we have demonstrated that **Embodied-R1 has established itself as the State-of-the-Art (SOTA) algorithm among Affordance/Modular-based VLAs.**
>
> Finally, we suspect this misunderstanding may stem from the inherent differences in decision-making paradigms. **We provide a detailed explanation of the advantages and limitations of Embodied-R1 and similar models in our response to the next question (Q7).**

---

> ### Author Response · Authors · 2025-11-25
> **Rebuttal for Reviewer LhSQ [4/5]**
>
> ## **Q7. A series of rebuttals regarding the improvements of Embodied-R1 in manipulation tasks**
>
> ### **A. Translation of Performance to Real-World Tasks**
>
> > However, how to prove such boost of performance will lead to improvements of real manipulation tasks?
>
>
>
> We emphasize that through extensive zero-shot generalization experiments, we have demonstrated that Embodied-R1 achieves a significant improvement in success rates compared to numerous strong baselines under fair settings. Since the error rate of the downstream planner remains constant across these comparisons, the results logically indicate that **the better Embodied-R1 is at translating task instructions into correct, executable pointing coordinates, the higher the success rate of the manipulation task.** This confirms that our performance gains directly translate to improvements in actual execution.
>
> ### **B. Comparison of Embodied Paradigms**
>
> > In other words, the object-level pointing paradigm is limited compared to the end-to-end VLA paradigm.
>
> We respectfully suggest that this concern may stem from a perspective difference regarding current embodied manipulation paradigms. As outlined in our Introduction, mainstream approaches generally fall into three categories: End-to-end VLA, Modular VLA, and Affordance VLA. We argue that the strengths and weaknesses of End-to-end VLAs versus Modular/Affordance VLAs are complementary rather than strictly superior/inferior:
>
> - **End-to-end VLAs offer high flexibility and can perform various tasks, but their current generalization capability is extremely limited:** State-of-the-art VLA models, such as the Pi series and Groot series, possess only very limited generalization and semantic understanding abilities. **The workflow for using VLA models involves three steps:** 1) Select a base model (e.g., Pi0); 2) Manually collect 50 demos per task (or hundreds for complex tasks) for a single task or a limited number of tasks; 3) Fine-tune the model using these data. Even so, models trained with additional SFT data can only achieve in-distribution generalization. Once the language instruction is rephrased, or the background or camera perspective changes slightly, the policy success rate drops significantly or even collapses entirely. When directly tested in zero-shot settings with new scenarios/camera perspectives/task instructions, these models show almost no understanding ability, **with success rates close to zero**. This is the core generalization dilemma of end-to-end VLAs, known as the "seeing-to-doing issue."
> - **Modular/Affordance VLAs excel in zero-shot generalization and can execute completely new tasks and scenarios:** Here, decision-making is divided into two components: perceptual understanding and planning. Modular VLAs typically leverage VLMs for understanding tasks, extract features via multiple visual models (e.g., SAM and DINO), and guide downstream planners. In contrast, Affordance VLAs use more direct multi-affordance cues to guide planners. In new scenarios and complex language instructions, Modular/Affordance VLAs still maintain a viable execution success rate.
>
> **Embodied-R1** is positioned as a high-performance Affordance VLA designed specifically for **zero-shot generalization**. Compared to strong baselines like FSD, RoboPoint, and MOKA, we have **achieved substantial improvements in benchmark performance** and task success rates. Even on **SIMPLEREnv**, a benchmark typically used for End-to-end VLA generalization, Embodied-R1 demonstrates top-tier performance.
>
> In summary, Embodied-R1’s core advantage is its powerful capability to understand diverse task instructions—including those requiring spatial reasoning, mathematical reasoning, and long-horizon planning(Fig.6), **without the need for the massive SFT required by end-to-end models. Its comprehensive pointing capabilities expand the range of executable tasks far beyond previous baselines.**

---

> ### Author Response · Authors · 2025-11-25
> **Rebuttal for Reviewer LhSQ [5/5]**
>
> ### **C. Applicability to Non-Object-Centric Tasks**
>
> Finally, regarding the limitation of Affordance/Modular VLAs in executing complex, non-object-centric tasks: We acknowledge that combining the strong semantic and spatial reasoning of Embodied-R1 with the flexibility of learning-based policies is a promising avenue.
>
> - **Expanded Experiments on LIBERO:** During the rebuttal, we significantly expanded the preliminary experiments in **Appendix F (Table 12, Page 22&23) in the revision**. We evaluated the integration of Embodied-R1 with a Diffusion Policy across all four suites of the LIBERO benchmark (Spatial, Object, Goal, and Long), rather than just the Spatial suite.
>
> |  | LIBERO Goal | LIBERO Spatial | LIbero Object | Libero Long | Avg |
> | --- | --- | --- | --- | --- | --- |
> | DP Baseline  | 64.3 | 76.1 | 71.4 | 45.9 | 64.4 |
> | DP + Embodied-R1 | **80.7** | **89.6** | **78.7** | **68.7** | **79.4 (↑15%)** |
>
> The results demonstrate that Embodied-R1 serves as an effective **Conditional Planner**. By feeding the generated Visual Trace as a condition to the downstream Diffusion Policy, we observed consistent performance gains. As shown in the table, the combined method (**DP + Embodied-R1**) achieved an average success rate of **79.4%**, significantly outperforming the standard Diffusion Policy baseline (**64.4%**).
>
> - **Future Work:** These results validate that Embodied-R1 can effectively guide learning-based policies. We plan to further explore deep integration with end-to-end VLAs, creating action inference models explicitly guided by these visual traces, combining the **generalization** of "pointing" with the **flexibility** of learning-based control.
>
> ---
>
> We hope these clarifications and our revisions have addressed your concerns. Thank you once again for your valuable time and expertise. We welcome any further questions or discussion.
>
> [1] Yuan et al. RoboPoint: A Vision-Language Model for Spatial Affordance Prediction for Robotics. CORL2024.
>
> [2] Yuan et al. From Seeing to Doing: Bridging Reasoning and Decision for Robotic Manipulation. Arxiv2025.
>
> [3] Ji et al. RoboBrain: A Unified Brain Model for Robotic Manipulation from Abstract to Concrete. CVPR2025.
>
> [4] Xu et al. A0: An Affordance-Aware Hierarchical Model for General Robotic Manipulation. ICCV2025.
>
> [5] Easy R1. https://github.com/hiyouga/EasyR1.
>
> [6] Wang et al. Affordance-R1: Reinforcement Learning for Generalizable Affordance Reasoning in Multimodal Large Language Model. Arxiv2025.
>
> [7] Fang et al. MOKA: Open-World Robotic Manipulation through Mark-Based Visual Prompting
>
> [8] Qi et al. SoFar: Language-Grounded Orientation Bridges Spatial Reasoning and Object Manipulation

---

### Official Review · Reviewer_4ftQ · 2025-10-31

**Soundness:** 3
**Presentation:** 4
**Contribution:** 3
**Rating:** 8
**Confidence:** 3

**Summary:**

This paper summarizes prior works with point-based affordances, redefines the concept of "pointing" as an intermediate representation, and extends it to four specific interpretations: Referring Expression Grounding (REG), Region Referring Grounding (RRG), Object Functional Grounding (OFG), and Visual Trace Generation (VTG). It collects and relabels datasets from multiple previous works, curates and re-annotates portions of them, specifically enhancing capabilities in these four areas to improve the accuracy of operations using pointing. The pointing format remains the classic xyz-coordinate point; however, by training on data with different semantic meanings, this representation acquires diverse types of knowledge. Downstream motion planners can more effectively utilize points enriched with these four types of information. To address the limitation of generalization when using supervised fine-tuning (SFT) with pointing, the paper employs GRPO for training.

**Strengths:**

1.	This paper systematically examines previous point-based works and explicitly categorizes the information required in embodied tasks into four distinct types. Although the representation itself is not particularly novel, the proposed classification scheme and the approach of constructing training data according to these four categories are relatively innovative.
2.	The work is solid and thorough, involving extensive data collection and cleaning, as well as comprehensive experimental validation.
3.	The application of RLVR methods in embodied AI is still rare. This paper takes a clever approach by not training end-to-end directly; instead, it uses reinforcement learning to train a perception module, and then passes the perception outputs to a motion planner for action execution.

**Weaknesses:**

1.	Baseline coverage could be broader. Although the authors compare to strong baselines like FSD and RoboPoint, some recent VLA models (e.g., VLA-RL, ThinkAct) are only cited but not experimentally compared, which slightly limits the comprehensiveness.
2.	Unclear differentiation from prior “point-based” or affordance representations. Although the paper repeatedly emphasises the novelty of the “pointing” representation, it does not clearly delineate how this differs conceptually and functionally from existing notions, such as affordance points, grasp points, or visual trajectory signals, used in prior works (e.g., RoboPoint, FSD). The reader is left uncertain whether “pointing” represents a new learning formulation, a different abstraction level, or simply a unified terminology. A more explicit comparison — e.g., in a figure or a dedicated subsection — between pointing, affordance, and trajectory signals would clarify the actual contribution.

**Questions:**

1.	The difference between the “pointing” representation and previous works should be further illustrated. For example, how do previous studies address the points and the issue of their dataset composition?
2.	It would be better for authors to add the evaluation results for all four LIBERO suites for further comparison. In the ablations, there is only the LIBERO-spatial suite.

---

> ### Author Response · Authors · 2025-11-25
> **Rebuttal for Reviewer 4ftQ [1/3]**
>
> We sincerely thank Reviewer 4ftQ for the positive assessment and for identifying the need to broaden our baseline coverage and clarify the conceptual boundaries of pointing. Below, we provide detailed responses and new experimental evidence. **All changes in the revised paper and supplementary materials have been highlighted in red for easy reference.**
>
> ### **Q1: Baseline Coverage (VLA-RL, ThinkAct, etc.).**
>
> We appreciate your suggestion to include more recent VLA models. We have expanded our comparisons in the revision.
>
> 1. **Qualitative Discussion (VLA-RL & ThinkAct vs. Embodied-R1): Vs. VLA-RL:** VLA-RL utilizes Reinforcement Learning primarily to fine-tune the policy for specific tasks. This approach typically requires extensive simulation interaction for each new task and lacks efficient cross-task transfer capabilities. In contrast, Embodied-R1 focuses on **zero-shot generalization** across diverse scenarios, tasks, and backgrounds by leveraging "Pointing" as a generalizable intermediate representation, without requiring task-specific RL fine-tuning or massive simulation data during deployment. **Vs. ThinkAct:** ThinkAct leverages latent thought processes to assist end-to-end decision-making. Our approach differs by using explicit reasoning to generate **precise, explicit pointing signals**. This explicit "pointing" output acts as an interpretable and verifiable bridge between high-level reasoning and low-level control.
> 2. **Quantitative Comparison in SIMPLEREnv:** To further expand the baseline coverage, **we have added ThinkAct and Magma to SIMPLEREnv** for comparison ****and updated more recent baseline algorithms **(pi0, pi0-fast, OpenVLA-OFT)**. we have updated **Section4.3 and** **Table 4 (page 8)** **in the revision** with new baselines. The detailed results are as follows:
>
> | Model | Type | Put Spoon on Towel | Put Carrot on Plate | Stack Green Block on Yellow Block | Put Eggplant in Yellow Basket  | Avg |
> | --- | --- | --- | --- | --- | --- | --- |
> | 𝜋0    | End-to-end VLA | 29.1 | 0.0  | 16.6  | 62.5  | 27.1 |
> | 𝜋0-fast | End-to-end VLA | 29.1 | 21.9  | 10.8 | 66.6 | 48.3 |
> | OpenVLA | End-to-end VLA | 4.2 | 0.0 | 0.0 | 16.7 | 5.2 |
> | OpenVLA-OFT | End-to-end VLA | 34.2 | 30.0 | 30.0 | **72.5** | 41.8 |
> | ThinkAct | End-to-end VLA | 58.3 | 37.5 | 8.7 | 70.8 | 43.8 |
> | Magma | End-to-end VLA | 37.5 | 31.0 | 12.7 | 60.5 | 35.4 |
> | MOKA | Modular VLA | 45.8 | 41.6  | 33.3 | 12.5  | 33.3 |
> | Sofar | Modular VLA | 55.5 | 56.9 | **62.5** | 40.2 | 53.8 |
> | RoboPoint  | Affordance VLA | 16.7 | 20.8 | 8.3 | 25.0 | 17.7 |
> | FSD | Affordance VLA | 41.6 | 50.0 | 33.3 | 37.5 | 40.6 |
> | Embodied-R1 | Affordance VLA | **62.5** | **68.0** | 36.1 | 58.3 | **56.2** |
>
> As shown in the updated table, Embodied-R1 achieves a **56.2%** success rate in the zero-shot OOD setting, outperforming **ThinkAct (43.8%)** and **Magma (35.4%)**. This demonstrates the superior generalization of our method compared to end-to-end baselines in novel environments.

---

> > ### Author Response · Authors · 2025-11-25
> > **Rebuttal for Reviewer 4ftQ [2/3]**
> >
> > ### **Q2: Differentiation from Prior Point-based Representations**
> >
> > We thank the reviewer for the suggestion to clarify the distinction between our work and prior arts. **In the Appendix H (page 25) of revision, we have specifically added a subsection to compare related work.**
> >
> > We conducted a comprehensive comparison with representative state-of-the-art baselines, including **RoboPoint** [1], **RoboBrain** [2], **A0** [3], and **FSD** [4]. The results are shown as follows:
> >
> > | Models | RoboPoint | RoboBrain | A0 | FSD | Embodied-R1 |
> > | --- | --- | --- | --- | --- | --- |
> > | Visual Aids Type | point | bounding box, robotic centric trace | object centric trace | bounding box, point, object centric trace | point, object centric trace |
> > | REG (Referring Expression Grounding) | Yes (use point) | Yes (use bounding box) | No | Yes (use bounding box and point) | Yes (use point) |
> > | RRG (Region Reference Grounding) | Yes  | No | No | Yes | Yes |
> > | OFG (Object Functional Grounding) | No | No | No | No | Yes |
> > | VTG (Visual Trace Generation) | No | Yes (robotic centric trace) | Yes (object centric trace) | Yes (object centric trace) | Yes (object centric trace) |
> > | Training Methods | SFT | SFT | SFT | SFT | RFT |
> > | Thinking | No | No | No | Yes (Fixed CoT) | Yes (Free-form Thinking) |
> > | Spatial/Logical Reasoning | N/A | N/A | N/A | Weak | Strong |
> >
> > The differences are fundamental across three key dimensions:
> >
> > **Observation 1: Unified Interface vs. Fragmented Capabilities.**
> > While "points" are a common primitive, prior methods utilize them in fragmented ways. A0 focuses exclusively on object-centric VTG but lacks the semantic understanding required for grounding (REG) or functional analysis (OFG). RoboPoint limits points to spatial constraints (REG, RRG). RoboBrain relies heavily on bounding boxes, which are often too coarse for precise manipulation (causing operation offsets), and utilizes robot-centric traces that limit cross-embodiment transfer. Embodied-R1 discards coarse bounding box signals. Instead, it innovatively unifies four core capabilities (REG, RRG, OFG, VTG) into a single, embodiment-agnostic point-based interface (where visual traces are sets of points). This unification allows Embodied-R1 to serve as a **"**Generalist Pointing Agent,**"** handling diverse manipulation needs within a single model.
> >
> > **Observation 2: Embodied-R1 acquires superior comprehension and reasoning abilities through RFT reasoning.** Baselines like FSD, RoboPoint, and A0 rely on SFT.
> >
> > - Models without a reasoning process (RoboPoint, A0) are limited to simple, explicit instructions (e.g., "point to the right of the plate") and fail at tasks requiring multi-step deduction (e.g., "pick up the object nearest to the robot").
> > - While FSD incorporates reasoning, it is constrained by fixed Chain-of-Thought (CoT) templates. As shown in our real-world experiments (Table 5 in the paper), FSD suffers from low success rates in tasks requiring spatial understanding or logical reasoning due to this rigidity in OOD scenarios.
> > - In contrast, Embodied-R1 is trained via RFT to develop free-form thinking and active spatial/logical reasoning. This allows it to handle complex instructions and reasoning-heavy tasks robustly. (See **Section 4.3 (Page 9) and Figure 6** in the revision for detailed visualization).
> >
> > **Observation 3: Multi-Task Synergy.**
> > By treating these distinct capabilities as a unified "Pointing" task during the RFT stage, with only 3B parameters, Embodied-R1 achieves superior coordinate understanding and semantic alignment. The massive performance gains observed across **10+ benchmarks** (spatial understanding and pointing) empirically validate that unifying these tasks under a reinforcement learning framework yields structural synergy that isolated pipelines lack.
> >
> > ### **Q3: Evaluation on All Four LIBERO Suites.**
> >
> > We have fully implemented your suggestion. We extended the integration experiment (using Embodied-R1's VTG to condition a Diffusion Policy) to all four LIBERO test suites. The new results are in **Appendix F (Table 12, Page 23) in the revision**.
> >
> > As shown in the table below, integrating Embodied-R1 leads to consistent performance gains across all suites. The combined method (**DP + Embodied-R1**) achieved an average success rate of **79.4%**, significantly outperforming the standard Diffusion Policy baseline (**64.4%**).
> >
> > | **Tasks** | **LIBERO Goal** | **LIBERO Spatial** | **LIBERO Object** | **LIBERO Long** | **Avg** |
> > | --- | --- | --- | --- | --- | --- |
> > | DP Baseline | 64.3 | 76.1 | 71.4 | 45.9 | 64.4 |
> > | **DP + Embodied-R1** | **80.7** | **89.6** | **78.7** | **68.7** | **79.4 (↑15%)** |
> >
> > These comprehensive results further validate Embodied-R1 as a robust, general-purpose intent generator for robotic manipulation.

---

> > > ### Author Response · Authors · 2025-11-25
> > > **Rebuttal for Reviewer 4ftQ [3/3]**
> > >
> > > ---
> > > We hope these clarifications and our revisions have addressed your concerns. Thank you once again for your valuable time and expertise. We welcome any further questions or discussion.
> > >
> > > [1] Yuan et al. RoboPoint: A Vision-Language Model for Spatial Affordance Prediction for Robotics. CORL2024.
> > >
> > > [2] Ji et al. RoboBrain: A Unified Brain Model for Robotic Manipulation from Abstract to Concrete. CVPR2025.
> > >
> > > [3] Xu et al. A0: An Affordance-Aware Hierarchical Model for General Robotic Manipulation. ICCV2025.
> > >
> > > [4] Yuan et al. From Seeing to Doing: Bridging Reasoning and Decision for Robotic Manipulation. Arxiv2025.

---

### Official Review · Reviewer_uNYS · 2025-10-31

**Soundness:** 4
**Presentation:** 4
**Contribution:** 4
**Rating:** 8
**Confidence:** 4

**Summary:**

This paper introduces Embodied-R1, a 3B Vision-Language Model (VLM) designed to bridge the "seeing-to-doing gap" in robotics, a problem it attributes to data scarcity and embodiment heterogeneity. The authors pioneer "pointing" as a unified, embodiment-agnostic intermediate representation to connect high-level comprehension with low-level actions. They define four core embodied pointing abilities—Referring Expression Grounding (REG), Region Referring Grounding (RRG), Object Functional Grounding (OFG), and Visual Trace Generation (VTG)—and introduce a large-scale dataset, Embodied-Points-200K, to train them. A key contribution is the training methodology: a two-stage Reinforced Fine-tuning (RFT) curriculum with a specialized multi-task reward system. This RFT approach is argued to be superior to standard Supervised Fine-Tuning (SFT) as it enables flexible, free-form reasoning and resolves the "multi-solution dilemma" (e.g., many valid points for an "empty space") inherent in pointing tasks. Embodied-R1 achieves state-of-the-art performance on 11 spatial and pointing benchmarks and demonstrates robust zero-shot generalization, achieving a 56.2% success rate in SIMPLEREnv and 87.5% across 8 real-world XArm tasks without task-specific fine-tuning, a 62% improvement over strong baselines. The model also exhibits high robustness to diverse visual disturbances.

**Strengths:**

- It pioneers "pointing" as a unified and embodiment-agnostic intermediate representation to bridge high-level vision-language understanding with low-level robot actions. This approach is designed to solve the "seeing-to-doing gap" by overcoming data scarcity and the challenge of hardware (embodiment) heterogeneity.

- The paper introduces Embodied-Points-200K, a large-scale, high-quality dataset curated from diverse sources to support four newly defined core pointing abilities: Referring Expression Grounding (REG), Region Referring Grounding (RRG), Object Functional Grounding (OFG), and Visual Trace Generation (VTG).

- It proposes a novel "Reinforced Fine-tuning (RFT)" training curriculum, which is a key advantage over standard Supervised Fine-Tuning (SFT). The RFT method enables flexible, free-form reasoning beyond rigid templates and effectively resolves the "multi-solution dilemma" by reinforcing any correct solution rather than overfitting to one.

- The resulting 3B parameter model, Embodied-R1, achieves state-of-the-art performance on 11 different spatial reasoning and embodied pointing benchmarks. It also demonstrates exceptional zero-shot generalization to robotic manipulation.

**Weaknesses:**

- The paper's core motivation for using Reinforced Fine-tuning (RFT) over Supervised Fine-tuning (SFT) is to address the "multi-solution dilemma" (e.g., many valid points for an "empty space") . However, an SFT loss can also be adapted to handle this by, for example, minimizing the distance to the closest valid point in a target mask rather than a single (x, y) coordinate. Could the authors clarify how the "Embodied-SFT" baseline was implemented for these ambiguous tasks? Does this baseline already account for the multi-solution problem, or does it naively overfit to a single sampled ground-truth point? This is critical for validating that RFT is a necessary solution to this specific problem, rather than just one possible solution.

- The paper pioneers "pointing" as an embodiment-agnostic representation. While this is effective for localization and trajectories, the authors acknowledge it may be insufficient for tasks requiring force, twisting, or interaction with deformable objects . How does the model currently handle instructions for which "pointing" is a fundamentally poor representation? For example, if given the command "unscrew the jar lid" or "wipe the table," would the model attempt to generate a point or trace, or would its reasoning process allow it to recognize the task's incompatibility with its action space?

- The preliminary results for the Embodied-R1-RGBD variant are concerning. The paper notes that on complex relational tasks, the 3D model performs worse than the 2D RGB-only model, a degradation attributed to potential "hallucinations" from the depth map . This seems to contradict the premise that richer, more relevant sensory information should improve performance. Could the authors elaborate on this finding? Does this suggest a fundamental flaw in the model's fusion architecture, where adding modalities can confuse the reasoning process, and how does this impact the future viability of extending this "pointing" paradigm into 3D?

- The paper's title claims "General Robotic Manipulation," yet the evaluations are limited to single-step instructions. The authors propose that a high-level planner could decompose long-horizon tasks as future work . While promising, this is a significant limitation. Could the authors provide any preliminary evidence, even a qualitative example, of Embodied-R1's ability to handle a simple, two-step instruction (e.g., "pick up the strawberry and then move it to the bowl") to demonstrate its feasibility as a robust execution module for such a hierarchical system?

- The authors astutely note that the automated pipeline for generating the Embodied-Points-200K dataset "inevitably introduces noise". How does the RFT training process handle this label noise compared to SFT? Is it more or less robust to an incorrect ground-truth trajectory or point? A brief analysis of the dataset's noise level and its effect on the reward signal would strengthen the paper's claims about the robustness of the RFT methodology.

**Questions:**

See the weakness section.

---

> ### Author Response · Authors · 2025-11-25
> **Rebuttal for Reviewer uNYS [1/3]**
>
> We sincerely thank Reviewer uNYS for the comprehensive assessment and the positive score. We are encouraged by your recognition of "Pointing" as a unified representation and the value of our RFT paradigm. Below, we provide detailed responses and new experimental evidence. **All changes in the revised paper and supplementary materials have been highlighted in red for easy reference.**
>
> ### **Q1: Motivation for RFT vs. SFT & Implementation of "Embodied-SFT".**
>
> Thank you for this deep inquiry into the core methodology.
>
> - **Embodied-SFT Implementation:** We confirm your understanding: the "Embodied-SFT" baseline utilizes a direct prediction loss fitting to a single coordinate. We adopted this formulation to align with the standard architecture of mainstream VLMs capable of pointing, such as **Molmo[1]** and **RoboPoint[2]**, which treat the task as predicting a deterministic point. We acknowledge that addressing the multi-solution nature of embodied tasks is critical. As you insightfully suggested, SFT can be adapted for this purpose, for instance, by using **reward-weighted SFT** to maximize likelihood over a masked region rather than a single point. While this is a promising direction, we maintained the vanilla SFT implementation as our baseline to benchmark against the most common existing methodology.
> - **Necessity of RFT:** While SFT can be adapted with weighted mask-based losses, RFT (via GRPO) offers structural advantages that are critical for solving the "multi-solution dilemma":
>     - **Exploration:** SFT inherently minimizes the divergence from the training distribution. In contrast, RFT incentivizes the model to explore *any* solution that yields a high reward. For instance, in a "place on the empty table" task, RFT learns to select any valid coordinate that maximizes success, rather than mimicking the average position of training demonstrations.
>     - **Grounding the Reasoning Chain:** A critical advantage of our RFT paradigm is the optimization of the intermediate reasoning process. The reward signal propagates back to update the internal monologue (`<think>...</think>`), effectively aligning the reasoning logic with the task goal. Unlike SFT, which only supervises the model to **mimic the text form** of reasoning, RFT reinforces the **correctness and causality** of the thought process based on execution feedback.
>     - **Unified Multi-Task Synergy:** RFT allows us to train diverse capabilities (Spatial QA, REG, VTG, etc) under a **unified reward formulation**, avoiding the complex manual tuning of loss weights required in multi-task SFT. Empirically, this leads to superior generalization across disparate tasks (Table 11).
>
> ### **Q2: How does the current model handle instructions for which "pointing" is essentially not applicable?**
>
> We appreciate you pointing out this limitation. Currently, Embodied-R1 is trained to generate a visual trace for any instruction. We agree that for impossible tasks, a "refusal" mechanism is ideal.
>
> At present, like Embodied-R1, end-to-end VLAs also face the issue of hallucinating actions when unable to execute tasks, continuing to act erroneously. We believe that future work should consider enhancing this capability, and implementation approaches can refer to error detection models such as AHA[3]. When tasks exceed the capabilities of the VLA, the system should refuse to execute them and raise an exception.
>
> ---
>
> **On the other hand, we also have a little surprise.** We believe pointing remains a powerful high-level signal even for contact-rich tasks. The semantic trajectory can guide a low-level force-aware controller. During the rebuttal, we validated this via two new tasks (**A detailed example of the task is presented in Section4.3 (Page 9) and Figure 6 in the revision**:):
>
> 1. **Wiping Blackboard:** Embodied-R1 generates a visual trace, which is executed by a low-level impedance controller. The visual trace defines where to wipe, while the controller handles the contact force.
> 2. **Press the bottle:** We combined Embodied-R1 (for visually locating buttons in varying positions) with a force-aware Diffusion Policy (for the continous pressing action). The system demonstrated strong generalization across different bottle positions.
>
> We aim to demonstrate that while Pointing inherently has certain representational limitations, it remains a flexible and under explored critical signal for robotic manipulation. Moreover, it serves as a viable and important pathway to bridge high-level perception with low-level decision-making, enabling cross-scenario generalization.

---

> ### Author Response · Authors · 2025-11-25
> **Rebuttal for Reviewer uNYS [2/3]**
>
> ### **Q3: Counter-intuitive Results on RGB-D Variant (3D < 2D).**
>
> Thank you very much for your excellent observation! We greatly appreciate your attention to some of our preliminary attempts and for providing extremely important suggestions that are worthy of further exploration.
>
> - **Initial Solution:** First, we introduce the scheme employed in the initial submission version. We first set the depth range to 0.4 meters to 2 meters based on the dataset scope, then normalize the depth data into a signal within the range of 0-1, convert it into a single-channel grayscale image, and subsequently duplicate it to form a three-channel depth map. Next, the RGB image and the depth map are input into the VLM as two separate images for processing and training. Since the same encoder is used, and the visual encoder of the original model is specifically designed for RGB images, there may be deficiencies in the alignment of depth maps, which could lead to hallucination issues. This indicates that we need to adopt a more sophisticated design to avoid modal conflicts, which we believe is a limitation of our initial scheme.
> - **New Architecture (Dual-Tower):** To address this, we redesigned the architecture during the rebuttal. Instead of concatenation, we adopted a **Dual-Tower structure** (inspired by *RoboRefer[4]*). One encoder processes RGB, and a separate encoder processes Depth.
> - **Training Strategy:** We implemented a two-stage training: (1) **Alignment:** SFT on the 1M RefSpatial (RGB-D) dataset to align the Depth encoder's features with the LLM space; (2) **RFT:** Full-parameter fine-tuning for 3D RRG tasks.
> - **New Results:** As shown in the updated table below, this new architecture (**Embodied-R1-RGBD-New**) significantly outperforms both the RGB baseline and the old RGB-D variant, particularly in complex relational tasks (Level 1).
>
> | Benchmark (Open6DOR) | Level 0 | **Level 1 (Complex Relations)** | **Overall** |
> | --- | --- | --- | --- |
> | Embodied-R1-RGB | 68.5 | 59.4 | 66.8 |
> | Embodied-R1-RGBD-Old | 99.8 | 50.9 | 90.2 |
> | **Embodied-R1-RGBD-New** | **99.2** | **74.5 (↑23.6%)** | **94.3** |
>
> This confirms the feasibility of 3D RRG tasks through proper feature alignment, thereby validating the effectiveness of our paradigm’s scalability to 3D. We believe that extending the pointing paradigm to more 3D tasks and broader scenarios still requires further design and adaptation, but its future feasibility is highly promising. This constitutes a key direction for our subsequent research. All results and analyses have been uploaded to **Appendix F, page 24** of the revised manuscript.
>
> ### **Q4: About Long-horizon Tasks**
>
> - **Clarification:** First, we would like to clarify that **multiple tasks demonstrated in our real-robot experiments have achieved the two-step instruction level you mentioned**. For instance, the instruction "pick up the strawberry and place it in the bowl" (as referenced by you) can be understood as a combination of the pick skill and the place skill. Tasks 2 to 7 are all such combinations of two-step skills; for example, "Move the nearest object to the right side of the drawer" not only involves two-step skills but also incorporates requirements for spatial reasoning capabilities.
> - **New Qualitative Examples:** We greatly appreciate your suggestions, we have truly integrated a high-level planner, which is capable of decomposing long-horizon tasks into multiple subtasks for execution. In the revision, we have added qualitative examples of **logic-driven and long-horizon tasks**, both of which demonstrate promising results. This further enhances the broad applicability of Embodied-R1 (see **Section4.3 (Page 9) and Figure 6 in the revision**):
> 1. **Reasoning Task:** "Place the sponge on the answer to the math problem on the whiteboard." (Requires logic reasoning + placement).
> 2. **Desktop Organization:** "Put all objects on the table into the plate." (Requires decomposing into multiple pick-and-place sub-tasks, planner: gemini-2.5-pro).
> 3. **Color-matched Duck Sorting:** "Place the three ducks onto plates of matching colors." (Requires iterative color matching and planning, planner:gemini-2.5-pro).
>
> These examples demonstrate Embodied-R1's capacity to handle temporal decomposition and logical reasoning, supporting its potential for general manipulation.

---

> > ### Author Response · Authors · 2025-11-25
> > **Rebuttal for Reviewer uNYS [3/3]**
> >
> > ### **Q5: Robustness to Label Noise (RFT vs. SFT)**
> >
> > Yes, we argue that RFT exhibits stronger robustness to label noise than SFT. If the labels are noisy (e.g., the target should be the object's center but is annotated at the edge, or the answer region is offset), SFT forces the model to memorize this error. With a large volume of such noise, the model will learn the average distribution of the noise. In contrast, for RFT, only part of the reward region may be erroneous, and extensive exploration reduces the probability of overfitting to a single incorrect point. Even if the reference labels contain noise, the model's behavior will be reinforced as long as it explores a more optimal point.
> >
> > On the other hand, ensuring high-quality data is a top priority in our pipeline construction. We have designed rigorous filtering and manual verification processes for each large-scale embodied dataset, rather than relying solely on automated scripts. For visual trace and functional point datasets, we implemented strict rule-based filtering and conducted **manual inspection** on subsets. We iteratively refined the filtering criteria until the accuracy of manual verification exceeded 95% before generating the complete dataset.
> >
> > ---
> >
> > We hope these clarifications and our revisions have addressed your concerns. Thank you once again for your valuable time and expertise. We welcome any further questions or discussion.
> >
> > [1] Deitke et al. Molmo and PixMo: Open Weights and Open Data for State-of-the-Art Vision-Language Models. CVPR2025.
> >
> > [2] Yuan et al. RoboPoint: A Vision-Language Model for Spatial Affordance Prediction for Robotics. CORL2024.
> >
> > [3] Duan et al. Aha: A vision-language-model for detecting and reasoning over failures in robotic manipulation. ICLR2025.
> >
> > [4] Zhou et al. RoboRefer: Towards Spatial Referring with Reasoning in Vision-Language Models for Robotics. NeurIPS2025.

---

> > > ### Comment · Reviewer_uNYS · 2025-11-26
> > >
> > > Thanks for the detailed rebuttal. I will keep my score.

---

### Official Review · Reviewer_gZir · 2025-10-31

**Soundness:** 4
**Presentation:** 4
**Contribution:** 3
**Rating:** 6
**Confidence:** 4

**Summary:**

This paper introduces Embodied-R1, a 3B Vision-Language Model (VLM) for general robotic manipulation. The core contribution is pioneering "pointing" as a unified, embodiment-agnostic intermediate representation to bridge the perception-action gap. The authors define four core embodied pointing abilities (REG, RRG, OFG, VTG) and train the model using a two-stage Reinforced Fine-tuning (RFT) curriculum on a custom large-scale dataset, Embodied-Points-200K. The RFT approach is crucial for handling the multi-solution nature of pointing tasks, which is superior to standard Supervised Fine-tuning (SFT). The model achieves state-of-the-art (SOTA) performance on spatial reasoning and strong zero-shot generalization in both simulation and real-world robot tasks.

**Strengths:**

- The work is original for two main reasons. First, the formal definition and systematization of four distinct embodied pointing abilities (REG, RRG, OFG, VTG) as a unified intermediate representation is a novel and intuitive approach to abstracting robot control. Second, the use of RFT specifically to address the multi-solution dilemma inherent in embodied pointing (e.g., many points are valid for an "empty space").
- The paper is written clearly and is easy to follow. The problem statement ("seeing-to-doing gap") is well-defined, and the proposed solution using "pointing" is logically introduced.
- The RFT-based, pointing-centric approach provides a new, highly effective, and scalable pathway for training VLMs for robotics, proving that a relatively smaller VLM (3B parameters) can outperform larger, specialized models by focusing on robust reasoning and generalization rather than just mimicking specific actions.

**Weaknesses:**

- Limited Exploration of Learning-Based Execution: The current main zero-shot robot evaluation (SIMPLEREnv and XArm) couples the VLM with a traditional CuRobo motion planner. While a promising preliminary experiment in Appendix F shows that the visual trace can boost a Diffusion Policy baseline, the core evaluation does not extensively explore this integration. Given that the paper emphasizes closing the "seeing-to-doing gap," relying predominantly on a classical planner might limit the realization of the full potential of the visual trace output, especially for complex or dynamic actions where a learned policy excels.
- Generalization Scope: The paper acknowledges that the "pointing" representation may be insufficient for complex manipulation requiring precise force control, wiping, or interaction with deformable objects. This limits the generality claim for the full spectrum of robotic manipulation.

**Questions:**

- 2D to 3D Trace Accuracy: The Visual Trace Generation (VTG) outputs a 2D trajectory which is then mapped to 3D using the pinhole camera model and initial depth information. How robust is this 2D-to-3D transformation process, particularly in the context of fine-grained manipulation tasks?
- I previously tested OpenVLA on the WidowX Robot in SIMPLEREnv using the same four tasks as in the paper, achieving an average success rate of around 40%. However, the paper reports 1.0%, which is a significant difference from 40%. Therefore, I am unsure whether this reported result is accurate.

---

> ### Author Response · Authors · 2025-11-25
> **Rebuttal for Reviewer gZir [1/3]**
>
> We sincerely thank Reviewer gZir for the detailed assessment and constructive feedback. Below, we provide detailed responses and new experimental evidence. **All changes in the revised paper and supplementary materials have been highlighted in red for easy reference.**
>
> ### **Q1: Limited Exploration of Learning-Based Execution.**
>
> We fully agree with your insight that integrating with learning-based methods is crucial for unleashing the full potential of our model, particularly in dynamic settings. We consider this a priority for both this rebuttal and future work.
>
> - **Expanded Experiments on LIBERO:** During the rebuttal, we significantly expanded the preliminary experiments in **Appendix F (Table 12, Page 22&23) in the revision**. We evaluated the integration of Embodied-R1 with a Diffusion Policy across all four suites of the LIBERO benchmark (Spatial, Object, Goal, and Long), rather than just the Spatial suite.
>
> |  | LIBERO Goal | LIBERO Spatial | LIbero Object | Libero Long | Avg |
> | --- | --- | --- | --- | --- | --- |
> | DP Baseline  | 64.3 | 76.1 | 71.4 | 45.9 | 64.4 |
> | DP + Embodied-R1 | **80.7** | **89.6** | **78.7** | **68.7** | **79.4 (↑15%)** |
>
> The results demonstrate that Embodied-R1 serves as an effective **Conditional Planner**. By feeding the generated Visual Trace as a condition to the downstream Diffusion Policy, we observed consistent performance gains. As shown in the table, the combined method (**DP + Embodied-R1**) achieved an average success rate of **79.4%**, significantly outperforming the standard Diffusion Policy baseline (**64.4%**).
>
> - **Future Work:** These results validate that Embodied-R1 can effectively guide learning-based policies. We plan to further explore deep integration with end-to-end VLAs, creating action inference models explicitly guided by these visual traces, combining the **generalization** of "pointing" with the **flexibility** of learning-based control.
>
> ### **Q2: Generalization Scope (Force Control and Complex Interactions).**
>
> We acknowledge that a purely "pointing" representation has inherent some limitations. However, we wish to clarify the capability boundary and our proposed solution:
>
> - **Role of Pointing:** "Pointing" and visual traces primarily resolve the **semantic and spatial planning** problem—determining "where to go" and "how to move". Precise force control typically operates at a lower level, requiring the execution of these spatial intents. Currently, even state-of-the-art **end-to-end VLAs struggle with zero-shot force control without extensive domain-specific demonstrations**.
> - **Hierarchical Compatibility:** Our framework is designed to be compatible with force-aware controllers. Embodied-R1 acts as a **"Generalization Plugin,"** providing high-level spatial guidance to a low-level controller (e.g., impedance control or a force-aware learned policy) that handles the real-time dynamics.
> - **New Exploratory Experiments:** During the rebuttal, we conducted qualitative analysis to validate this approach in **Section4.3 (Page 9) and Figure 6 in the revision**:
>     - **Wiping Blackboard:** We utilized the Embodied-R1 to generate a coverage path (wiping trajectory) and fed this into a low-level impedance controller. This combination successfully performed the wiping task by maintaining contact force while following the semantic trajectory.
>     - **Press the bottle:** We combined Embodied-R1 (for visually locating buttons in varying positions) with a force-aware Diffusion Policy (for the pressing action softly). The system demonstrated strong generalization across different bottle positions.
>
> We aim to demonstrate that while Pointing inherently has certain representational limitations, it remains a flexible and under explored critical signal for robotic manipulation. Moreover, it serves as a viable and important pathway to bridge high-level perception with low-level decision-making, enabling cross-scenario generalization.

---

> ### Author Response · Authors · 2025-11-25
> **Rebuttal for Reviewer gZir [2/3]**
>
> ### **Q3: Robustness of 2D to 3D Trajectory Transformation.**
>
> We address the robustness of the 2D-to-3D projection from two perspectives:
> 1. **Semantic Stability:** Embodied-R1 is trained to predict points on semantically significant and geometrically stable features (e.g., object centers, handles). The depth values at these specific regions are generally continuous and reliable. Furthermore, our VTG module applies trajectory smoothing when execution, which effectively suppresses noise from single-pixel depth errors.
> 2. **Real-World Stress Testing:** Our real-world experiments (Table 5 & Figure 7 in the paper) serve as a rigorous stress test. We utilized a single Intel RealSense L515 camera, introducing inherent challenges such as depth noise, occlusion during execution, and imperfect calibration. Despite these physical imperfections, Embodied-R1 achieved an **87.5% success rate** across 8 diverse tasks. This empirical evidence strongly suggests that the system is robust to the levels of geometric noise encountered in practical deployments.
>
> ### **Q4: Discrepancy in OpenVLA Performance in SIMPLEREnv.**
>
> Thank you for highlighting this discrepancy; ensuring fair baselines is critical.
> In our initial submission, all the end-to-end VLAs results (including OpenVLA) were **referenced from the *SpatialVLA[1]* paper.** Therefore, OpenVLA only achieves 1% performance.
> During the rebuttal period, we carefully reviewed the results and **reproduced the performance of OpenVLA in SIMPLEREnv, obtaining an average of 5.2%**. We also cross-checked the results with those reported in several recent papers, such as ThinkAct[2] and InternVLA-M1[3], and found them to be similar. However, we noticed that when **reproducing OpenVLA-OFT[4] on the WidowX task, a performance of 41.8% could be achieved, consistent with your observations.**
>
> Additionally, to provide a comprehensive view, we have updated **Section4.3 and** **Table 4 (page 8)** **in the revision** with new baselines, including OpenVLA-OFT, ThinkAct, and Magma[5]. The detailed results are shown below:
>
> | Model | Type | Put Spoon on Towel | Put Carrot on Plate | Stack Green Block on Yellow Block | Put Eggplant in Yellow Basket  | Avg |
> | --- | --- | --- | --- | --- | --- | --- |
> | 𝜋0    | End-to-end VLA | 29.1 | 0.0  | 16.6  | 62.5  | 27.1 |
> | 𝜋0-fast | End-to-end VLA | 29.1 | 21.9  | 10.8 | 66.6 | 48.3 |
> | OpenVLA | End-to-end VLA | 4.2 | 0.0 | 0.0 | 16.7 | 5.2 |
> | OpenVLA-OFT | End-to-end VLA | 34.2 | 30.0 | 30.0 | **72.5** | 41.8 |
> | ThinkAct | End-to-end VLA | 58.3 | 37.5 | 8.7 | 70.8 | 43.8 |
> | Magma | End-to-end VLA | 37.5 | 31.0 | 12.7 | 60.5 | 35.4 |
> | MOKA | Modular VLA | 45.8 | 41.6  | 33.3 | 12.5  | 33.3 |
> | Sofar | Modular VLA | 55.5 | 56.9 | **62.5** | 40.2 | 53.8 |
> | RoboPoint  | Affordance VLA | 16.7 | 20.8 | 8.3 | 25.0 | 17.7 |
> | FSD | Affordance VLA | 41.6 | 50.0 | 33.3 | 37.5 | 40.6 |
> | Embodied-R1 | Affordance VLA | **62.5** | **68.0** | 36.1 | 58.3 | **56.2** |
>
> Even against these strengthened baselines, Embodied-R1 demonstrates superior zero-shot generalization (56.2%), significantly outperforming the base OpenVLA in OOD settings and surpassing the finetuned OpenVLA-OFT and all Modular VLAs.
>
> ### **Additional Qualitative Examples**
>
> To demonstrate the broad applicability of Embodied-R1, we conducted additional experiments on complex tasks, evaluating two distinct categories as shown in **Section4.3 (Page 9) and Figure 6 in the revision**:
>
> - **Long-Horizon Tasks (Planner-Assisted):** For tasks like *"Sort colored ducks"* and *"Desktop organization,"* which require temporal decomposition, we integrated Embodied-R1 with a high-level planner (Gemini-2.5-pro). The planner breaks down the instruction into iterative sub-goals, and Embodied-R1 provides precise spatial grounding for each step to execute the sequence.
> - **Reasoning Tasks (Model-Native):** For logic-driven tasks such as *"Place the sponge on the answer to the math problem,"* the process relies on the model's intrinsic reasoning capabilities. Embodied-R1 performs logical deduction to identify the semantic target derived from the calculation directly, ensuring accurate localization.
>
> These results validate that Embodied-R1 acts as a robust low-level grounding module that can be seamlessly integrated with LLM planners to handle long-horizon tasks, while also possessing the internal logic required for reasoning-heavy scenarios.

---

> > ### Author Response · Authors · 2025-11-25
> > **Rebuttal for Reviewer gZir [3/3]**
> >
> > We hope these clarifications and our revisions have addressed your concerns. Thank you once again for your valuable time and expertise. We welcome any further questions or discussion.
> >
> > [1] Qu et al. Spatialvla: Exploring spatial representations for visual-language-action model. RSS2025.
> >
> > [2] Huang et al. Thinkact: Vision-language-action reasoning via reinforced visual latent planning. 2025. NeurIPS2025.
> >
> > [3] Chen et al. InternVLA-M1: A Spatially Guided Vision-Language-Action Framework for Generalist Robot Policy. Arxiv2025.
> >
> > [4] Kim et al. Fine-Tuning Vision-Language-Action Models: Optimizing Speed and Success. RSS2025.
> >
> > [5] Yang et al. Magma: A Foundation Model for Multimodal AI Agents. CVPR2025.

---

### Meta-Review · Area_Chair_j6ZN · 2026-01-10

**Summary:**

The reviewers generally acknowledge the merit of Embodied-R1 in pioneering "pointing" as a unified, embodiment-agnostic intermediate representation. Three of the four reviewers (gZir, uNYS, 4ftQ) gave positive scores, highlighting the innovative use of Reinforcement Fine-Tuning (RFT) to resolve the "multi-solution dilemma" inherent in spatial tasks. The model’s efficiency—achieving state-of-the-art (SOTA) performance across 11 benchmarks with only 3B parameters—was also highly praised.

However, several key concerns informed the final decision process:


+ Taxonomy and Terminology: Reviewer LhSQ strongly contested the initial classification of the model as a "Vision-Language-Action (VLA)" model, arguing it is a perception model that does not output actions directly. In response, the authors clarified the architectural distinctions and updated the terminology to "Affordance-based Methods".


+ Originality and Concurrent Work: A significant outlier concern involved the paper's similarity to concurrent RFT-based vision works (e.g., Seg-Zero, VisionReasoner) and allegations of high code similarity. The authors successfully rebutted these by explaining the shared use of the EasyR1 open-source framework and demonstrating the uniqueness of their robotics-domain dataset and reward functions.


+ Generalization and Execution: Some reviewers questioned the scope of generalization for contact-rich or long-horizon tasks. The rebuttal addressed this by providing new experiments on the full LIBERO benchmark suite, showing a 15% success rate boost when integrated with a Diffusion Policy.


+ 3D Capability: Initial concerns regarding the poor performance of the RGB-D variant were resolved through the introduction of a new Dual-Tower Architecture, which significantly improved 3D spatial reasoning accuracy on the Open6DOR benchmark.

**Reviewer Concerns:**

**Concerns Addressed by the Rebuttal:**

+ Baseline Coverage and Evaluation: The authors successfully expanded their comparative analysis in Section 4.3 and Table 4 to include recent baselines such as ThinkAct, Magma, and $\pi_0$-fast, confirming the model's superior zero-shot performance.

+ Integration with Learning-based Policies: In response to Reviewer gZir, the authors provided extensive results on the full LIBERO benchmark suite, demonstrating that using Embodied-R1 as a conditional planner for a Diffusion Policy yields a 15% improvement in success rates.

+ 3D Perception and Architecture: Addressing Reviewer uNYS’s concerns regarding the RGB-D variant, the authors introduced a new Dual-Tower architecture with a modality alignment phase, significantly improving 3D spatial reasoning accuracy on the Open6DOR benchmark to 94.3%.

+ Terminology and Task Scope: The authors addressed the taxonomic concerns raised by Reviewer LhSQ by relabeling the model as an "Affordance-based method" and providing qualitative evidence for its effectiveness in long-horizon and reasoning-heavy tasks, such as color-matched duck sorting and math-based localization.


**Outstanding Concerns:**

+ Fundamental Methodological Novelty: Reviewer LhSQ maintains a firm stance that the innovation is limited, as the model architecture and training objective (GRPO-based RFT) are highly similar to existing works like VisionReasoner and Seg-Zero, with the primary difference residing in the domain-specific data and reward functions.

+ Scalability Analysis: Concerns regarding how the model’s performance scales beyond the 3B parameter magnitude were not explored through additional experiments, as the authors prioritized deployment efficiency on real-world hardware.


+ Inherent Limitations of Pointing: While the model is robust for geometric planning, its sufficiency for the "full spectrum" of manipulation—specifically tasks requiring force control, twisting, or interaction with deformable objects—remains an acknowledged representational boundary.

**Reviewer Scores:**

Reviewer gZir (Initial: 6): The reviewer would likely have increased their score to a 7 or 8. All technical concerns—including the reproduction discrepancy of OpenVLA, baseline coverage with recent models like ThinkAct, and the robustness of the 2D-to-3D projection—were addressed with substantial new data and clarifications.


Reviewer uNYS (Initial: 8): This reviewer's score would remain a stable 8. They explicitly acknowledged that the rebuttal provided detailed counterarguments and successfully demonstrated a fix for the RGB-D performance degradation through the new Dual-Tower architecture.


Reviewer 4ftQ (Initial: 8): The score would likely stay at an 8. The authors fulfilled every specific request, including adding a comprehensive conceptual comparison section in the appendix and providing evaluation results for all four suites of the LIBERO benchmark.


Reviewer LhSQ (Initial: 2): Based on the final discussion, this reviewer would likely have raised their score to a 4 or 5. While they still maintained that the methodological novelty was "insufficient" due to similarities with prior RFT works, they explicitly stated a willingness to "appropriately raise" their score after the authors accepted the taxonomy correction and added missing citations.

---

### Decision · Program_Chairs · 2026-01-26

Accept (Poster)